

# FjordRPM v1.0: a reduced-physics model for efficient simulation of glacial fjords

Donald A. Slater[1], Eleanor Johnstone[1], Martim Mas e Braga[2], Neil Fraser[3], Tom Cowton[2], and Mark Inall[3]

[1]School of Geosciences, University of Edinburgh, Edinburgh, UK
[2]School of Geography and Sustainable Development, University of St Andrews, St Andrews, UK
[3]Scottish Association for Marine Science, Oban, UK

**Correspondence:** Donald Slater (donald.slater@ed.ac.uk)

**Abstract.** Interactions between ice masses and the ocean are key couplings in the global climate system. In many cases these interactions occur through glacial fjords, which are long, deep, and narrow troughs connecting the open ocean to marine-terminating glaciers. By controlling the fluxes of ocean heat towards the ice sheet and ice sheet freshwater towards the ocean, glacial fjords play an important role in modulating ice sheet mass loss and the impacts of freshwater on ocean circulation.
Yet, these dynamics occur at small scales that are challenging to resolve in earth system models and so are they often ignored, represented in an ad-hoc manner, or studied using expensive high-resolution models that are limited in scope. Here, we propose a means of capturing glacial fjord dynamics at negligible computational expense in the form of a "reduced-physics" model (FjordRPM) that resembles a "1.5-dimensional" or box model. We describe the design and physical parameterisations in the model and demonstrate its ability to capture important modes of glacial fjord circulation by comparing it against a general circulation model in idealised and realistic simulations. We suggest that the model is a useful tool for understanding fjord dynamics and a promising approach for representing glacial fjord processes within large-scale models or climate and sea level projection efforts.

## 1 Introduction

There is increasing recognition that ice-ocean interaction is a key exchange in our earth system. Ocean heat is linked to ice sheet mass loss and sea level contribution (Straneo and Heimbach, 2013), ice sheet freshwater has the potential to influence large-scale ocean dynamics (Böning et al., 2016) and ice-ocean processes fertilise ecosystems (Oliver et al., 2023). One prominent system in which these interactions occur is glacial fjords, which link marine-terminating glaciers to the open ocean. Glacial fjords are found in glaciated regions around the world such as the West Antarctic Peninsula, Svalbard, and Alaska, and are particularly numerous in Greenland. Around two-thirds of the mass loss from the Greenland Ice Sheet since the 1970s has occurred at marine-terminating glaciers that flow into glacial fjords (Mouginot et al., 2019), and there is concern that the associated increased freshwater flux to the ocean may be capable of weakening the Atlantic Meridional Overturning Circulation (Frajka-Williams et al., 2016; Jackson et al., 2023; van Westen et al., 2024). As such, the ability to capture the effects of glacial fjord dynamics in models used for climate and ice sheet projections is crucial.





Glacial fjords, with a typical width of 2-10 km, are however much too small to resolve in earth system or global ocean
models and generally too small to include in even regional ocean models (e.g., Zuo et al., 2019; Nguyen et al., 2021). When
considering the impact of the ocean on the Greenland Ice Sheet in large-scale simulations (e.g., Goelzer et al., 2020), glacial
fjord dynamics have, therefore, either been ignored or represented in a very basic manner (Slater et al., 2020), missing important
details of how subglacial discharge, icebergs, shelf winds and sills all modify the properties of ocean waters reaching marine-
terminating glaciers (Mortensen et al., 2014; Jackson et al., 2014; Straneo and Cenedese, 2015; Hager et al., 2022; Davison
et al., 2020). Similarly, when considering the impact of Greenland Ice Sheet freshwater on ocean dynamics, models often
impose the freshwater as an unmixed and distributed surface flux (e.g., Golledge et al., 2019) or make an a-priori choice of
depth-distribution (e.g., Gillard et al., 2016), neither of which is faithful to our understanding of how ice sheet liquid freshwater
enters the ocean as a highly diluted, subsurface flux that can be delayed in time (Beaird et al., 2018; Sanchez et al., 2023) and
of how a significant portion of the solid ice flux melts inside fjords (Moon et al., 2018; Moyer et al., 2019). In sum, better
representation of glacial fjord processes within large-scale studies is needed to fully understand ice-ocean interactions and
their impact on ice masses and climate.

Here, we introduce a means of approximating glacial fjord dynamics at negligible computational cost, which we name
FjordRPM ("Fjord Reduced Physics Model"). The model describes the volume-mean fjord temperature and salinity in a number
of discrete, fixed, vertically-stacked layers. Exchange processes between the fjord, glacier, shelf and icebergs are parameterised
using known or adapted expressions. The resulting model lies somewhere between a box model and a "1.5-dimensional" model
(i.e., a 1-dimensional model with lateral fluxes) and hence we adopt the terminology "reduced physics model". Models of this
type have a history of providing insight into ocean systems (e.g., Babson et al., 2006; Gillibrand et al., 2013) and of representing
coastal processes within larger-scale ocean models (e.g., Sun et al., 2017). It is our hope that FjordRPM can be of similar use
within the field of ice sheet-ocean interaction.

The paper proceeds as follows. We first describe the formulation of FjordRPM and the parameterisation of exchanges
(section 2), followed by the method of solution, the structure of the code and the required inputs (section 3). We demonstrate
that the model can capture fundamental modes of glacial fjord circulation by comparing to simulations conducted using the
established general circulation model MITgcm (section 4). We conclude by reflecting on the strengths and weaknesses of the
model and proposing avenues for future work (section 5).

## 2 Model description

### 2.1 Preliminaries

Glacial fjords are host to a range of processes that serve to influence a fjord's circulation and properties (e.g., Straneo and
Cenedese, 2015). In this first iteration of FjordRPM, we focus on a small number of key processes which we deem to be of
primary importance (Fig. 1). Two of these are specific to glacial environments and are recognised to be of particular significance
in glacial fjords. Firstly, the input of subglacial discharge at the glacier grounding line drives buoyant plumes that mix with and
upwell ambient waters before reaching the fjord surface or finding neutral buoyancy at an intermediate depth (Jenkins, 2011;





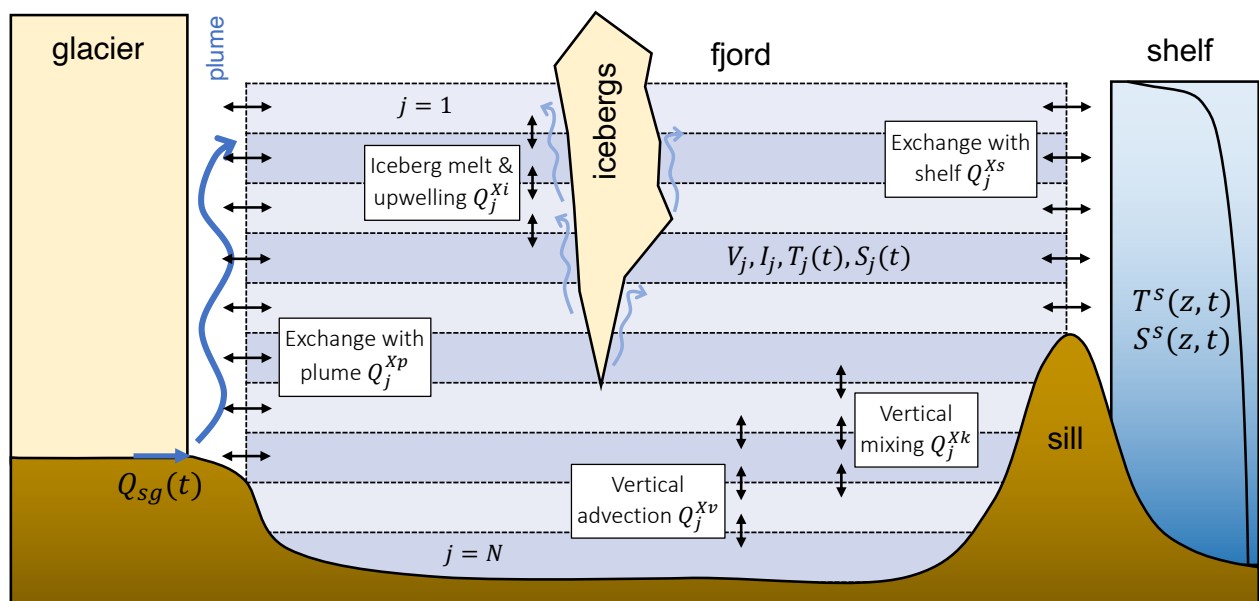

**Figure 1.** Schematic of FjordRPM. The model layers ($j$) have the static properties of volume, $V_j$, iceberg surface area, $I_j$, and the dynamic properties of temperature, $T_j(t)$, and salinity, $S_j(t)$. The layers have horizontal exchanges of volume ($X = V$, heat ($X = T$) and salt ($X = S$) with the plume ($Q_j^{Xp}$) and the shelf ($Q_j^{Xs}$). They have vertical exchanges due to iceberg melt and upwelling ($Q_j^{Xi}$), vertical advection ($Q_j^{Xv}$) and vertical mixing ($Q_j^{Xk}$). The forcing boundary conditions are the subglacial discharge, $Q_{sg}(t)$, and the shelf temperature, $T^s(z,t)$, and salinity, $S^s(z,t)$.

Jackson et al., 2017; Hewitt, 2020). Secondly, the melting of icebergs calved into the fjord from tidewater glaciers acts as a heat sink and freshwater source whilst driving convective upwelling adjacent to icebergs (Moon et al., 2018; Davison et al., 2020; Cenedese and Straneo, 2023). In addition to these processes, we include the exchange of water between the fjord and
60 continental shelf, as driven by pressure gradients in response to spatial and temporal variation in water properties (e.g., Jackson et al., 2014), and vertical mixing and advection between model layers.

We split the fjord (Fig. 1) into $N$ vertically-stacked layers, indexed by $j$ from the surface ($j = 1$) to the bottom ($j = N$). Each layer has a volume $V_j$, which is fixed in time and which may be converted to a layer thickness $H_j$ by taking into account the hypsometry of the fjord. At present, the model assumes a cuboid-shaped fjord of length $L$ and width $W$, so that $V_j = WLH_j$.
65 Each layer also has a static iceberg area in contact with the ocean $I_j$, and a time-varying temperature $T_j(t)$ and salinity $S_j(t)$ that can be considered as an average over the volume of the layer. The number of layers and their volumes are user-defined; the layers do not have to have equal volume or thickness, though the sum of the layer thicknesses has to equal the fjord depth. Fewer layers means less vertical resolution while more layers limits the time step of the model. If the fjord has a sill (e.g., Fig. 1)





then the model will slightly adjust the thicknesses to ensure that the sill depth coincides with a layer boundary, simplifying the
formulation of the fjord-shelf exchange.

The layers experience time-dependent exchanges (Fig. 1) with the subglacial discharge-driven plumes (entrainment and intrusion), with the shelf (fjord-shelf exchange), with icebergs (melting and upwelling) and with adjacent layers (vertical mixing and vertical advection). Note that the model allows for multiple plumes since multiple marine-terminating glaciers can terminate in a single fjord. Other glacial fjord exchange processes and drivers of circulation, such as ambient melting of the glacier, surface fluxes, sea ice, winds and tides were considered lower priority and are not yet implemented, but this is not intended to imply that these processes are not important in some situations. Exchange fluxes are denoted by $Q_j^{Xy}$, where $j$ denotes the layer where the exchange is happening, $X$ denotes the flux quantity being exchanged ($V$ for volume, $T$ for heat or $S$ for salt) and $y$ denotes the process driving exchange ($p$ for plume, $s$ for shelf, $i$ for icebergs, $k$ for vertical mixing or $v$ for vertical advection).

For fluxes that represent exchange between adjacent layers (iceberg-driven upwelling, vertical mixing and vertical advection), a further line of explanation is required. These fluxes naturally take a general form $Q_{j+1,j}^{Xy}$, where $j+1$ and $j$ are the layers exchanging. These fluxes can, however, be written in the single-subscript format previously introduced by noting that the net effect on layer $j$ is the difference between the exchange with layer $j+1$ and the exchange with layer $j-1$; that is, $Q_j^{Xy} = Q_{j+1,j}^{Xy} - Q_{j,j-1}^{Xy}$. In this expression, there is no exchange between the bottom layer and the sea floor, or between the surface layer and atmosphere ($Q_{N+1,N}^{Xy} = Q_{1,0}^{Xy} = 0$).

For sign convention, a volume flux $Q_j^{Vy}$ is defined as positive if it is adding volume to a layer. A volume flux between layers $Q_{j+1,j}^{Vy}$ is defined as positive if it removes volume from the deeper layer $j+1$ and adds it to the shallower layer $j$. To give some examples, $Q_2^{Vp}$ is the volume exchange between the plume and layer 2, positive if the plume is adding volume to layer 2, and $Q_5^{Vs}$ is the volume exchange between fjord layer 5 and the shelf, positive if it is directed from the shelf into the fjord.

Estimating many of the exchanges requires knowledge of the density differences between two water masses. To do this, we employ a linear equation of state, such that the relative buoyancy of two water masses $a$ and $b$ is defined by

$$g'_{ab} = g \left[ \beta_S \left( S_a - S_b \right) - \beta_T \left( T_a - T_b \right) \right] \tag{1}$$

in which $g$ is the gravitational acceleration and $\beta_S$ and $\beta_T$ are the haline contraction and thermal expansion coefficients, respectively.

We now describe in turn how we parameterise exchange with the plume (section 2.2), with the shelf (section 2.3), with icebergs (section 2.4), due to vertical mixing (section 2.5) and due to vertical advection (section 2.6). For reference, a full list of model variables and parameters, including their units, is given in Tables A1 & A2.

## 2.2 Exchange with the plume

If there is subglacial discharge emerging from beneath a marine-terminating glacier, this will drive an upwelling plume adjacent to the glacier that entrains water at depth and intrudes into the fjord at a level of neutral buoyancy (Cowton et al., 2015; Carroll et al., 2015; Stevens et al., 2016; Mankoff et al., 2016). Within FjordRPM this is represented by a flux leaving the deeper layers





and a flux entering a shallower layer (Fig. 1). To quantify these fluxes, we employ a now-standard approach that has been termed "plume-melt theory" by Jackson et al. (2022), consisting of the classic theory for buoyant plumes (Morton et al., 1956) coupled to the standard three-equation parameterisation for submarine melt (Holland and Jenkins, 1999). If there are multiple

plumes emerging from a single glacier, or multiple glaciers with plumes draining into a single fjord, the following approach is applied separately for each plume.

We assume a line plume geometry (Jackson et al., 2017) with a fixed plume width in the across-fjord direction of $W_p$. As the plume rises, plume volume, momentum, heat and salt flux per unit width evolve according to

$$\frac{d}{dz}\left(b_p u_p\right) = \alpha_p u_p + m_p \tag{2a}$$

$$\frac{d}{dz}\left(b_p u_p^2\right) = b_p g_p' - C_d u_p^2 \tag{2b}$$

$$\frac{d}{dz}\left(b_p u_p T_p\right) = \alpha_p u_p T + m_p T_b - C_d^{1/2} \Gamma_T u_p \left(T_p - T_b\right) \tag{2c}$$

$$\frac{d}{dz}\left(b_p u_p S_p\right) = \alpha_p u_p S + m_p S_b - C_d^{1/2} \Gamma_S u_p \left(S_p - S_b\right) \tag{2d}$$

in which $b_p$ is plume width in the along-fjord direction, $u_p$ is plume (vertical) velocity, $T_p$ is plume temperature, $S_p$ is plume salinity, $\alpha_p$ is the plume entrainment coefficient, $g_p' = g\left[\beta_S\left(S - S_p\right) - \beta_T\left(T - T_p\right)\right]$ is the relative buoyancy of the fjord and

plume, and $T$ and $S$ are the fjord temperature and salinity (see also Table A1). All other variables and parameters relate to the submarine melt rate induced by the plume, $m_p$, which is given by three equations that balance heat and salt flux through the ice-ocean boundary layer and maintain the boundary layer at the in-situ freezing point (e.g., Jenkins, 2011)

$$m_p l + m_p c_i \left(T_b - T_i\right) = c_w C_d^{1/2} \Gamma_T u_p \left(T_p - T_b\right) \tag{3a}$$

$$m_p S_b = C_d^{1/2} \Gamma_S u_p \left(S_p - S_b\right) \tag{3b}$$

$$T_b = \lambda_1 S_b + \lambda_2 + \lambda_3 |z| \tag{3c}$$

where $T_b$ and $S_b$ are the temperature and salinity of the ice-ocean boundary layer, $l$ is the latent heat of ice melt, $c_i$ and $c_w$ are the heat capacities of ice and seawater, $C_d$ is the plume-ice drag coefficient, $\Gamma_T$ and $\Gamma_S$ are heat and salt transfer coefficients and $\lambda_i$ are the constants that control the linearised freezing point (see also Table A2).

The boundary conditions for Eqs. 2 are the fjord temperature $T$ and salinity $S$, which are taken from the FjordRPM layers.

The plume is initiated by a flux of subglacial discharge $Q_{sg}$ emerging from beneath the glacier at the grounding line depth $H_{gl}$, with initial salinity $S_{sg} = 0$ and temperature $T_{sg} = \lambda_2 + \lambda_3 H_{gl}$ (i.e., the in-situ freezing point). The initial plume width and velocity are set by ensuring that the plume has a balance of buoyancy and momentum (e.g., Slater et al., 2016) and an initial flux of $Q_{sg}$, giving $b_p = \left(\alpha_p Q_{sg}^2 / g_p' W_p^2\right)^{1/3}$ and $u_p = Q_{sg}/b_p W_p$.

At each FjordRPM timestep, Eqs. 2 & 3 are numerically integrated using an Euler scheme on the vertical grid defined by the

interfaces between the FjordRPM model layers (Fig. 1). To provide an example based on Eq. 2a, if the plume width, velocity and melt rate at the interface between layers $j+1$ and $j$ are $b_p^{j+1,j}$, $u_p^{j+1,j}$ and $m_p^{j+1,j}$, then the plume volume flux at the next interface up, between layers $j$ and $j-1$, is estimated by

$$b_p^{j,j-1} u_p^{j,j-1} = b_p^{j+1,j} u_p^{j+1,j} + H_j \left(\alpha_p u_p^{j+1,j} + m_p^{j+1,j}\right) \tag{4}$$





Beginning from the grounding line, this numerical integration continues to shallower layers until the plume finds neutral
buoyancy and intrudes into the fjord. The layer of plume neutral buoyancy is defined as the layer where the plume-fjord
relative buoyancy, $g_p'$, switches from positive to negative, indicating that the plume is negatively buoyant in that layer and
will stop rising and intrude horizontally. If the plume reaches the fjord surface without becoming negatively buoyant then it
intrudes into the surface layer. In this manner, we obtain the plume velocity $u_p$ and submarine melt rate $m_p$ at each of the layer
interfaces.

The volume flux per unit plume width entrained from the fjord into the plume is the first term on the right hand side of
Eq. 2a, while the associated heat and salt fluxes are the first terms on the right hand side of Eqs. 2c & d. For layers where the
plume is rising, the exchange fluxes between the model layers and the plume (Fig. 1) are therefore given by

$$Q_j^{V_p} = -\alpha_p W_p u_p^{j+1,j} H_j \tag{5a}$$
$$Q_j^{T_p} = -\alpha_p W_p u_p^{j+1,j} H_j T_j \tag{5b}$$
$$Q_j^{S_p} = -\alpha_p W_p u_p^{j+1,j} H_j S_j \tag{5c}$$

where we know $u_p^{j+1,j}$ from the numerical integration. For any layers that are deeper than the grounding line, or shallower
than the layer where the plume intrudes into the fjord, all plume fluxes are 0.

In the layer where the plume intrudes into the fjord, say $j_0$, the flux from the plume into the fjord consists of the subglacial
discharge, the submarine meltwater, and all of the water that has been entrained into the plume. Thus, the volume flux from the
plume into the layer of neutral buoyancy is

$$Q_{j_0}^{V_p} = Q_{sg} + W_p \sum_j H_j \left[ \alpha_p u_p^{j+1,j} + m_p^{j+1,j} \right] \tag{6}$$

where the sum runs over all layers $j$ between the grounding line and the layer of neutral buoyancy. To obtain the heat flux
exchange, we apply the same principle, but we first use Eq. 3a to rewrite Eq. 2c as

$$\frac{d}{dz} (b_p u_p T_p) = \alpha_p u_p T + m_p T_{eff} \tag{7}$$

where

$$T_{eff} = T_b - \frac{l}{c_w} - \frac{c_i}{c_w} (T_b - T_i) \tag{8}$$

is the effective meltwater temperature (e.g., Jenkins, 2011). Noting the relative size of the terms, we furthermore approximate
$T_{eff} \approx -l/c_w$. The heat flux from the plume into the layer of neutral buoyancy can then be calculated as

$$Q_{j_0}^{T_p} = Q_{sg} T_{sg} + W_p \sum_j H_j \left[ \alpha_p u_p^{j+1,j} T_j + m_p^{j+1,j} T_{eff} \right] \tag{9}$$

Lastly, the salt flux from the plume into the layer of neutral buoyancy can be calculated as

$$Q_{j_0}^{S_p} = W_p \sum_j H_j \left[ \alpha_p u_p^{j+1,j} S_j \right] \tag{10}$$



where we have used the fact that the salinity of subglacial discharge, $S_{sg}$, is zero, and that the last two terms on the right hand side of Eq. 2d cancel according to Eq. 3b. The sums in Eqs. 9 & 10 again run over all layers $j$ between the grounding line and layer of neutral buoyancy.

By these definitions, the plume removes water from the deeper layers, mixes it with subglacial discharge and submarine meltwater and puts the resulting mixture back into a layer closer to the surface, representing the upwelling and transformation of fjord waters by the plume and driving an overturning circulation. While the plume exchange fluxes in a given layer may be large, note that since all of the volume flux entrained from deeper layers is put back into a shallower layer, the net volume flux over all layers associated with the plume is simply the subglacial discharge and the submarine melt flux; that is

$$\sum_{j=1}^{N} Q_j^{V_p} = Q_{sg} + \sum_{j=1}^{N} W_p H_j m_p^{j+1,j} \equiv Q_{sg} + Q_{sm} \tag{11}$$

Similarly, all of the heat and salt entrained from deeper layers is put back into a shallower layer, and the net heat and salt flux associated with the plume come only from the subglacial discharge and submarine melting.

## 2.3    Exchange with the shelf

We assume that exchange between the fjord and shelf is driven by pressure gradients between the fjord and shelf. Previous
studies have considered cases where: (i) this pressure gradient is balanced by friction and mixing (Geyer and MacCready, 2014; Sanchez et al., 2023); or (ii) is in geostrophic balance; or (iii) under hydraulic control (Zhao et al., 2021). The relevant balance for a given fjord will depend on the fjord and sill geometry, the stratification and the circulation. We have derived our exchange parameterisation with regime (i) in mind, but we note that it is not incompatible with regime (ii).

In setting up such balances, we neglect an acceleration term in the momentum balance, meaning that FjordRPM will not
resolve the transient response to high-frequency shelf variability. This may result in a model more suited to studying the fjord response to monthly or seasonal variability in environmental forcing, a point we return to in the results and discussion.

For layers that are below the sill (Fig. 1) there is no direct exchange between that layer and the shelf. For layers that are above the sill, the relative buoyancy between the shelf and fjord in layer $j$ is

$$g'_{sj} = g \left[ \beta_S \left( S_j^s - S_j \right) - \beta_T \left( T_j^s - T_j \right) \right] \tag{12}$$

where $S_j^s$ and $T_j^s$ are the shelf temperature and salinity profiles mapped onto the FjordRPM grid (Fig. 1). That is,

$$S_j^s(t) = \frac{1}{H_j} \int_j S^s(z,t) \, dz \tag{13}$$

and the integral runs over the depth range covered by layer $j$. An equivalent expression applies for temperature. Now define a quantity $\phi$, given for each layer by

$$\phi_j = g'_{sj} H_j / 2 + \sum_{k=1}^{j-1} g'_{sk} H_k \tag{14}$$




Since $\phi_j$ accumulates the shelf-fjord buoyancy difference over all layers shallower than $j$, it follows that the pressure difference between layer $j$ and the same depth range on the shelf is then given by $\rho_0\phi_j$, where $\rho_0$ is a reference density. We then define the fjord-shelf volume exchange fluxes for above-sill layers as

$$Q_j^{Vs} = WH_j\left(u_b + C_0\frac{\phi_j}{L}\right) \tag{15}$$

where $WH_j$ is the cross-sectional area of contact between layer $j$ and the shelf, $u_b$ is a barotropic velocity that ensures overall conservation of fjord volume (see below) and $\phi_j/L$ is essentially the fjord-shelf pressure gradient. $C_0$ controls the strength of the exchange and accounts for the fjord-shelf balance that is being assumed. We do not dig further into this balance; rather we treat $C_0$ as an empirical constant but return to this point in the discussion.

The associated heat and salt fluxes are obtained by noting that if, for layer $j$, the volume flux is directed out of the fjord, then the salinity associated with the volume flux should be that of the layer, while if the volume flux is directed into the fjord then the salinity should be that of the shelf. Thus, the salt fluxes associated with exchange with the shelf are given by

$$Q_j^{Ss} = \begin{cases} Q_j^{Vs}S_j & \text{if } Q_j^{Vs} < 0 \\ Q_j^{Vs}S_j^s & \text{if } Q_j^{Vs} \geq 0 \end{cases} \tag{16}$$

with similar expressions for heat.

Regarding the barotropic velocity $u_b$, note that if the sum of all the volume fluxes into the fjord is not zero, there will be a change in total fjord volume. This could be accounted for by dynamically tracking the sea surface height of the fjord relative to the shelf and its impact on fjord to shelf pressure gradients. However, the adjustment timescale for the sea surface height is very fast (on the order of minutes), and therefore tracking sea surface height requires a very small model timestep that limits the rest of the model. In addition, the sea surface height rapidly approaches a steady state in which the fjord-shelf volume fluxes have adjusted to ensure there is no net volume flux into the fjord. On the timescales of the model timestep (hours to days), it is therefore a very good approximation to choose $u_b$ in Eq. 15 to ensure there is no net volume flux into the fjord. This is equivalent to the 'rigid lid' approach sometimes applied in general circulation models. Other than the shelf fluxes, the only contribution to net volume change is from the plume fluxes, where the net volume flux comprises the subglacial discharge and submarine meltwater and is given by Eq. 11. Thus, if there are $N_{above}$ above-sill layers, conservation of overall fjord volume means

$$Q_{sg} + Q_{sm} + \sum_{j=1}^{N_{above}} Q_j^{Vs} = 0 \tag{17}$$

from which, combining with Eq. 15, we can obtain the barotropic velocity, $u_b$, as

$$u_b = -\frac{1}{W\sum_{j=1}^{N_{above}} H_j}\left(Q_{sg} + Q_{sm} + \frac{WC_0}{L}\sum_{j=1}^{N_{above}} H_j\phi_j\right) \tag{18}$$

which completes the definition of the shelf fluxes.





## 2.4 Exchange with icebergs

Iceberg melting freshens and cools fjords (Hager et al., 2024; Abib et al., 2024), and the input of the buoyant meltwater gives
melt-driven convective plumes rising up the sides of icebergs (Cenedese and Straneo, 2023). These plumes could be represented
by plume-melt theory (Magorrian and Wells, 2016), as for the subglacial discharge plumes in section 2.2, but we would require
many such plumes and would have to decide on the basis of iceberg characteristics what depth to initiate these plumes. We
are not convinced that such complexity is merited within a reduced-physics model and so we have aimed to take a simpler
approach without losing the key physics.

Starting with iceberg melting, numerous parameterisations exist for iceberg melt rate (e.g., Neshyba and Josberger, 1980;
FitzMaurice et al., 2017; Cenedese and Straneo, 2023). Ideally, we would use the three equation melt rate parameterisation
(Eqs. 3), however it is not straightforward to estimate the relative velocity of the icebergs and ocean that would be required. We
therefore instead take the simple approach of assuming that the melt rate of icebergs in layer $j$ is proportional to the thermal
forcing of that layer, with a constant of proportionality $M_0$. Once scaled for the layer iceberg surface area, $I_j$, the iceberg melt
flux is given by

$$Q_j^{melt} = M_0 \left( T_j - T_j^f \right) I_j \tag{19}$$

where $M_0$ is a constant and $T_j^f = \lambda_1 S_j + \lambda_2 + \lambda_3 z_j$ is the linearised in-situ freezing point, which depends on the layer salinity
$S_j$ and the mean depth of the layer, $z_j = H_j/2 + \sum_{n=1}^{j-1} H_n$.

   To estimate the resulting upwelling flux, we follow scalings from Magorrian and Wells (2016) for melt-driven convection.
Leaving the details to Appendix A, the scale of the (vertical) convection velocity resulting from iceberg melting in layer $j$ can
be taken to be

$$v_j = \left[ \frac{Q_j^{melt} g'_{j,melt} H_j}{\alpha_i I_j} \right]^{1/3} \tag{20}$$

where $\alpha_i$ is the entrainment coefficient and $g'_{j,melt}$ is the relative buoyancy of the layer and meltwater, given by

$$g'_{j,melt} = g \left[ \beta_S S_j - \beta_T \left( T_j - T_{eff} \right) \right] \tag{21}$$

in which $T_{eff} = -l/c_w$ is the effective meltwater temperature as in section 2.2.

   By making an analogy with a line plume of width $W_j = I_j/H_j$, the entrainment flux into the rising melt-driven convection
plume is then

$$Q_j^{ent} = \alpha_i v_j H_j W_j = \alpha_i v_j I_j \tag{22}$$

In a stratified fjord, however, the melt-driven convection cell will reach neutral buoyancy some height after initiating. The
length scale that controls this height can be estimated, again following scalings in Magorrian and Wells (2016) and detailed in
Appendix A, by

$$l_{j+1,j}^{ice} = \frac{v_{j+1}^2}{H_{j+1}} \frac{H_j + H_{j+1}}{2g'_{j+1,j}} \tag{23}$$





If $l^{ice}_{j+1,j} > H_{j+1}$, then any melt-driven convection cell that initiates in layer $j+1$ will upwell to layer $j$. If, however, $l^{ice}_{j+1,j} = H_{j+1}/2$ (say), then only melt-driven convection cells initiating in the upper half of layer $j+1$ would upwell over the layer

boundary to layer $j$. Assuming there is an equal probability of melt-driven convection cells initiating at any point in the layer, the resulting upwelling flux would be half of that in Eq. 22. The final iceberg upwelling volume flux from layer $j+1$ to $j$ is thus given by Eq. 22 scaled by a fraction $f^{ice}$ that accounts for how far iceberg meltwater is able to upwell

$$Q^{Vi}_{j+1,j} = \min\left[1, \frac{l^{ice}_{j+1,j}}{H_{j+1}}\right] Q^{ent}_{j+1} \equiv f^{ice}_{j+1,j} Q^{ent}_{j+1} \tag{24}$$

Note that $f^{ice}_{N+1,N} = f^{ice}_{1,0} = 0$ because there is no upwelling into the bottom layer or from the surface layer. The associated

temperature and salt fluxes are given by multiplying by the temperature and salinity of the layer from which the volume is upwelling.

Having accounted for the upwelling driven by iceberg melting, we return to the melting itself. The impact of melting ice on ambient ocean waters is equivalent to adding freshwater of effective temperature $T_{eff}$ to the ambient (Jenkins, 2011). In addition, the volumes of iceberg meltwater are much smaller than the layer volumes, so that we may treat the meltwater as a

virtual flux, meaning that we account for its impact on layer temperature and salinity but not on layer volume. Under these assumptions, and if all of the iceberg meltwater in layer $j$ stays in layer $j$, then the layer heat flux associated with the iceberg melt flux $Q^{melt}_j$ is $-Q^{melt}_j l/c_w$ and the salt flux is $-Q^{melt}_j S_j$.

Putting it all together, the net volume flux into layer $j$ due to iceberg melting is

$$Q^{Vi}_j = Q^{Vi}_{j+1,j} - Q^{Vi}_{j,j-1} \tag{25}$$

where the terms on the right hand side are given by Eq. 24. The net heat flux into layer $j$ is

$$Q^{Ti}_j = Q^{Vi}_{j+1,j} T_{j+1} - Q^{Vi}_{j,j-1} T_j - f^{ice}_{j+1,j} Q^{melt}_{j+1} \frac{l}{c_w} - \left(1 - f^{ice}_{j,j-1}\right) Q^{melt}_j \frac{l}{c_w} \tag{26}$$

where the first term is the upwelling of ambient water from the layer below, the second term is the upwelling of ambient water to the layer above, the third term is the upwelling of meltwater from the layer below, and the fourth term is the meltwater generated in the present layer that stays in the present layer. Finally, the net salt flux into layer $j$ is

$$Q^{Si}_j = Q^{Vi}_{j+1,j} S_{j+1} - Q^{Vi}_{j,j-1} S_j - f^{ice}_{j+1,j} Q^{melt}_{j+1} S_j - \left(1 - f^{ice}_{j,j-1}\right) Q^{melt}_j S_j \tag{27}$$

Note that by definition, $\sum_{j=1}^{N} Q^{Vi}_j = 0$, so that there is no fjord-scale volume flux associated with iceberg melting. In doing the same sum for heat and salt flux, the first two terms on the right hand sides of Eqs. 26 & 27 cancel because upwelling vertically redistributes heat and salt but does not change the fjord-scale content. The second two terms on the right hand sides of Eqs. 26 & 27 do not however cancel, giving a net cooling and freshening due to iceberg melting.

## 2.5 Vertical mixing

Exchange between layers occurs due to vertical mixing at the layer interfaces. We assume that this causes an exchange of heat and salt but no net exchange of volume, and calculate the vertical turbulent tracer fluxes as the product of an eddy diffusivity,



$K_z$, and the vertical gradient of the tracer. Once scaled for the area of contact between layers $j+1$ and $j$, this gives a salt exchange of

$$Q^{Sk}_{j+1,j} = W L K_z \frac{\partial S}{\partial z} = 2 W L K_z \frac{S_{j+1} - S_j}{H_{j+1} + H_j} \tag{28}$$

and an equivalent expression applies for temperature (we make no distinction between the eddy diffusivity of heat and salt). This eddy diffusivity is estimated as a function of the local Richardson number, Ri, using the KPP scheme (Large et al., 1994)

$$K_z = K_b + \begin{cases} K_0 & \text{Ri} \le 0 \\ K_0 \left[ 1 - (\text{Ri}/\text{Ri}_0)^2 \right]^3 & \text{Ri}_0 > \text{Ri} > 0 \\ 0 & \text{Ri} \ge \text{Ri}_0 \end{cases} \tag{29}$$

where $\text{Ri}_0 = 0.7$ is the Richardson number above which shear-driven mixing is suppressed, $K_0 = 5 \times 10^{-3}\,\mathrm{m^2\,s^{-1}}$ scales the shear-driven mixing and $K_b = 1 \times 10^{-5}\,\mathrm{m^2\,s^{-1}}$ is the background vertical mixing associated with internal waves (Table A2; Large et al., 1994). The Richardson number takes the usual definition $\text{Ri} = N^2/(\partial u/\partial z)^2$ and is estimated at the interface between layers $j+1$ and $j$ as

$$\text{Ri} = \frac{2 g'_{j+1,j}/(H_{j+1} + H_j)}{\left[2 \left(u_{j+1} - u_j\right)/\left(H_{j+1} + H_j\right)\right]^2} = \frac{g'_{j+1,j}\left(H_{j+1} + H_j\right)}{2 \left(u_{j+1} - u_j\right)^2} \tag{30}$$

where $g'_{j+1,j}$ is the relative buoyancy of the two layers according to Eq. 1. Lastly, the horizontal velocity scale for each layer, $u_j$, is estimated using the plume and shelf exchange fluxes as

$$u_j = \frac{Q^{Vp}_j - Q^{Vs}_j}{2 W H_j} \tag{31}$$

By these definitions, tracer vertical mixing is inhibited by stratification between layers, but enhanced by velocity shear driven by the exchange between layers and the plume or shelf. Like for the iceberg fluxes, these layer-to-layer tracer fluxes can be converted into net fluxes for each layer using

$$Q^{Sk}_j = Q^{Sk}_{j+1,j} - Q^{Sk}_{j,j-1} \tag{32}$$

and an equivalent expression applies for net heat fluxes.

## 2.6 Conservation of layer volume

By the described parameterisations, there are volume fluxes into or out of the layers due to the plume, shelf and icebergs. If, for a given layer, these fluxes do not sum to zero, then in a model with fixed layer volumes there must be an additional volume flux that ensures conservation of layer volume. By analogy with models that calculate vertical velocities by imposing incompressibility, we take this additional volume flux to be vertical advection between layers. As an illustrative example, consider a fjord with a plume and a shallow sill that significantly restricts the inflow and outflow. The water that is upwelled by the plume is not able to be replaced directly (i.e., horizontally) by inflow from the shelf. There must therefore be downwelling





(or reflux) within the fjord to replace the water entrained into the plume. This downwelling would be represented by the vertical
flux now described.

Suppose that the net volume flux imbalance in layer $j$ by the parameterisations described so far is $Q_j^{imb}$, given by

$$Q_j^{imb} = Q_j^{Vp} + Q_j^{Vs} + Q_j^{Vi} \tag{33}$$

For the surface layer ($j = 1$), and since there can be no volume flux to the atmosphere, we ensure conservation of volume by
imposing a flux from the second layer into the surface layer given by $Q_{2,1}^{Vv} = -Q_1^{imb}$. To ensure conservation of volume for the
second layer, we must then impose a flux from the third layer into the second layer given by $Q_{3,2}^{Vv} = -Q_2^{imb} + Q_{2,1}^{Vv}$. We can
continue iterating to get the required vertical fluxes for all boxes with the general expression

$$Q_{j+1,j}^{Vv} = -\sum_{n=1}^{j} Q_n^{imb} \tag{34}$$

The associated heat and salt fluxes depend on the direction of the flux. For salt flux, this is written as

$$Q_{j+1,j}^{Sv} = \begin{cases} Q_{j+1,j}^{Vv} S_{j+1} & \text{if } Q_{j+1,j}^{Vv} \geq 0 \\ Q_{j+1,j}^{Vv} S_j & \text{if } Q_{j+1,j}^{Vv} < 0 \end{cases} \tag{35}$$

That is, if the flux is directed from layer $j+1$ to $j$ then the relevant salinity is $S_{j+1}$, whereas if the flux is directed from layer
$j$ to $j+1$ the relevant salinity is $S_j$. As for icebergs and vertical mixing, these layer-to-layer fluxes can finally be converted to
net fluxes for each layer using

$$Q_j^{Sv} = Q_{j+1,j}^{Sv} - Q_{j,j-1}^{Sv} \tag{36}$$

with equivalent expressions applying for volume and heat fluxes.

## 2.7 Evolution equations

Having now defined all of the required fluxes (Fig. 1), we turn to the equations giving the time evolution of the model. Since
layer volumes are fixed in time, the only evolution equations are for the salt and heat content of each layer. For layer salt
content we have

$$V_j \frac{dS_j}{dt} = Q_j^{Sp} + Q_j^{Ss} + Q_j^{Si} + Q_j^{Sk} + Q_j^{Sv} \tag{37}$$

where the terms are, respectively, the salt fluxes associated with the plume(s) (Eqs. 5c & 10), shelf (Eq. 16), icebergs (Eq. 27),
vertical mixing (Eq. 32) and vertical advection (Eq. 36). For layer heat content we have the equivalent

$$V_j \frac{dT_j}{dt} = Q_j^{Tp} + Q_j^{Ts} + Q_j^{Ti} + Q_j^{Tk} + Q_j^{Tv} \tag{38}$$

These evolution equations, together with all of the flux definitions throughout sections 2.2-2.6, form a closed set of equations
that can be numerically integrated in time.





## 2.8 Density inversions

Before describing the numerics of FjordRPM, we note one last way in which layer temperature and salinity can evolve, which is that we enforce total mixing of two adjacent layers if the upper layer becomes denser than the lower layer. Physically, this represents convection. Specifically, at each time step and for each pair of adjacent layers $j+1$ and $j$ we check the buoyancy jump $g'_{j+1,j}$ between the layers using Eq. 1. For any pair of layers having $g'_{j+1,j} < 0$ we reset the salinity of the layers to

$$S_{j+1} = S_j = \frac{S_{j+1}V_{j+1} + S_jV_j}{V_{j+1} + V_j} \tag{39}$$

with an equivalent equation for temperature.

## 3 Numerical implementation

### 3.1 Code structure and method of solution

FjordRPM is coded in MATLAB, though future plans include a translation to an open source language such as Python or Fortran. We make use of a number of structures to efficiently carry around the required variables: 'p' contains all the model parameters, 'f' contains the forcing boundary conditions, 'a' contains the initial conditions and 's' contains the solution. The time vector for the model is denoted 't'. Full lists of the variables in these structures is included for reference in Tables A1 & A2. The code is arranged in a modular fashion (Fig. 2) and proceeds as follows. The top level routine (*run_model*), takes as input the structures p, t, f and a, and outputs the solution s. Within this routine, the code first makes a number of checks on the inputs (*check_inputs*), such as ensuring they have the correct dimensions. The solution variables are initialised and the initial conditions read in (*initialise_variables*). The forcing boundary conditions may be provided on a depth and time grid that differs from the model layers and time stepping, so some spatial and temporal interpolation of the forcings may be required (*bin_forcings*).

The model then enters the main time-stepping loop (Fig. 2). At each time step $t_i$, with dynamic variables $T_j(t_i)$ and $S_j(t_i)$, the first action is to check for and resolve any density inversions (*homogenise_unstable_layers*; section 2.8). The code then uses the temperature and salinity of the layers at the current time step $[T_j(t_i), S_j(t_i)]$, together with all of the parameters and variables that do not vary in time, to calculate the heat and salt fluxes described in sections 2.2-2.6 (*compute_fluxes* and functions therein). With these fluxes, the model makes a forward Euler time step of Eqs. 37 & 38 to obtain the temperature and salinity $[T_j(t_{i+1}), S_j(t_{i+1})]$ of the layers at the next time step (*step_solution_forwards*). The final stage of the time-stepping loop is to check that the sum of the volume flux exchange for each layer, which should be 0, is below a certain tolerance (*check_solution*). Once the time-stepping loop has finished, a final function (*get_final_output*) processes the solution into a final form for saving as output.

A special note is required here for the plume fluxes. Variability in the plume dynamics is determined by the subglacial discharge and fjord water properties. Since subglacial discharge and fjord water properties will generally vary only on timescales of days, while the model time step may be several hours, it is not essential that we 'renew' the plume dynamics every single





```
run_model % top level routine
      check_inputs % checking everything has been provided correctly
      initialise_variables % set up the simulation
            bin_forcings % get the forcings on model layers and time steps
      for i=1 to number of time steps % time-stepping loop
            homogenise_unstable_layers % resolve any density inversions
            compute_fluxes % calculate all the required layer fluxes
                  get_plume_fluxes % between layers and plume
                  get_shelf_fluxes % between layers and shelf
                  get_mixing_fluxes % vertical mixing between layers
                  get_iceberg_fluxes % icebergs
                  get_vertical_fluxes % vertical advection between layers
            step_solution_forwards % make the time step
            check_solution % check error tolerance
      end % end time-stepping loop
      get_final_output % finalise model output to be saved
```

**Figure 2.** Schematic of FjordRPM code.

time step. Indeed, integrating the melt-plume theory (Eqs. 2 & 3) is one of the slowest parts of the model and so updating the plume dynamics only every so often can substantially speed up the model. To implement this, note that the exchange fluxes between the plume and fjord (Eqs. 5, 6, 9 & 10) depend on the plume velocity, the induced submarine melt rate and the fjord properties. Thus, we can use the same plume velocity and induced submarine melt rate for a number of time steps, while using the fjord properties from the current time step. Essentially, this decouples the dynamics of the plume from the fjord properties for a number of time steps, but maintains conservation of heat and salt because the temperature and salinity that enters the plume fluxes is always from the current time step. The parameter *run_plume_every* defines the number of time steps between updates of the plume dynamics.

## 3.2 Required inputs

As inputs, the model first requires specification of all of the physical parameters contained in the structure 'p'. Default values for all of these parameters are provided in a function *default_parameters*; some, like the heat capacity of seawater $c_w$, are unlikely to need changed while others, such as the shelf exchange parameter $C_0$, are more likely to be varied by the user. The geometry of the problem must also be specified, including the length, width and depth of the fjord, the presence and depth of a possible sill, and the grounding line depth of any glaciers with plumes.

The user must also specify the number and thickness of the model layers. The layers do not all have to have the same thickness, so the user could choose to have higher vertical resolution at depths of interest. A large number of layers gives the solution better vertical resolution, but will require a shorter time step (see next section) and increase the computation time, though this is not normally an issue unless doing an ensemble of hundreds of simulations. We recommend roughly 20-80 layers, giving a vertical resolution of $O(10\,\text{m})$ in Greenland's fjords. When there is a sill, the model may make a small adjustment to





380 the layer thicknesses (in the *initialise_variables* routine) to ensure that the sill depth lies at a layer boundary, simplifying the calculation of the shelf fluxes.

The forcing boundary conditions must be provided in the structure 'f'. These include the subglacial discharge for each plume as a function of time, and the shelf temperature and salinity as a function of depth and time. Finally, the initial conditions must be specified in the structure 'a'. These include the initial temperature and salinity of the layers, and also the surface area of 385 icebergs in each layer. The latter is, for now, classified as an initial condition rather than a forcing because it does not vary in time.

### 3.3  Stability of time-stepping

The last required input is the time vector, 't', encompassing the duration and time step for the simulation. Like most models, a time step that is too large can lead to numerical instability. The complexity of the right hand side of Eqs. 37 & 38, however, 390 makes it difficult to derive a general limit on the stable time step. If we took into account only the mixing fluxes, Eq. 38 becomes a 1D heat equation with a spatially and temporally-variable diffusivity. With a forward Euler scheme, as employed here, stable time steps of the 1D heat equation must satisfy $\Delta t \leq (\Delta z)^2 / 2K_z$, where $\Delta z$ would be our layer thickness and $K_z$ is the eddy diffusivity of heat. With $K_z$ at a reasonable maximum possible value ($5 \times 10^{-3}$ m$^2$ s$^{-1}$, section 2.5) and $\Delta z = 10$ m, the maximum stable time step is around 0.1 days. However, in FjordRPM this condition guarantees neither stability nor 395 instability, because $K_z$ is a function of the Richardson number and only reaches such a high value in a few layers and/or at a few time steps, and because vertical mixing is just one of the total 5 fluxes.

Another handle on stability is the fraction of a layer volume that is exchanged in a given time step. For example, if a vertical volume flux is $Q$ (section 2.6) then the fraction of layer volume $V$ that is exchanged in a time step $\Delta t$ due to this flux is $Q\Delta t/V$. Writing the volume flux as $Q = WLv$, with $v$ the vertical velocity, and the volume as $V = WL\Delta z$, with $\Delta z$ the 400 layer thickness, the fraction of layer volume exchanged in a time step can be expressed as $v\Delta t/\Delta z$, which is an advective CFL condition. Investigations have shown that instability is often associated with this fraction exceeding $\sim 0.5$; and with a typical vertical velocity scale of $10^{-4}$ m s$^{-1}$ and a layer thickness $\Delta z = 10$ m this gives a maximum stable time step of around 0.5 days. But again this condition is not always relevant, because instability can also arise from horizontal fluxes or the vertical mixing, or interactions between these fluxes. Furthermore, we do not have these fluxes in advance of running the simulation so 405 this does not provide an easy way of choosing the time step.

In practice, the model is sufficiently cheap to run that the pragmatic way to choose the time step is to try a number of values and check the solution for instability. As a guide, we have found that a stable time step is on the order of 0.1-4 days for realistic forcings and layer thicknesses of 10s m. Such simulations take no more than a few seconds per model year on a laptop.

### 3.4  Character of the model

410 Before moving onto example results, we provide a few insights into the character of the model. Although there are many parameters and possible choices, the key dynamics boil down to just a handful of parameters - essentially one per exchange flux process.



The plume fluxes depend principally on the subglacial discharge $Q_{sg}$ and plume width $W_p$. For a given subglacial discharge, the main 'adjustable' parameter is the plume width. A smaller value will decrease the flux of plume waters that are input to the layer at neutral buoyancy and this neutral buoyancy layer will tend to be closer to the surface. Conversely a larger value of plume width will increase the flux of plume waters and the neutral buoyancy level will tend to be deeper.

The only adjustable parameter in the shelf fluxes is the constant $C_0$, which determines the strength of the fjord-shelf exchange for a given fjord-shelf pressure gradient. Large values of $C_0$ give quick communication between fjord and shelf. Small values of $C_0$ inhibit fjord-shelf exchange, meaning that, for example, plume waters struggle to exit the fjord and there is recirculation and retention within the fjord. This tends to vertically-redistribute waters in the fjord, leading to fjord-shelf exchange that is weaker and more distributed in the vertical compared to that obtained with a large value of $C_0$.

Within the iceberg fluxes the main adjustable parameter is $M_0$, which controls the iceberg melt rate for a given temperature. For a given layer temperature, an increase in $M_0$ will give higher iceberg melt rate and therefore higher iceberg melt flux. This has a greater cooling and freshening effect on fjord waters, will drive greater upwelling of fjord waters by icebergs, and will in turn drive greater fjord-shelf exchange.

Lastly, in the vertical mixing the main adjustable parameter is $K_0$, which controls the magnitude of vertical mixing of heat and salt. A higher value smoothes out vertical gradients in temperature and salinity, which can then contribute to the vertical structure of fjord-shelf exchange. There are no parameters associated with the vertical advective fluxes since these are calculated as a residual of all the other fluxes.

In summary, for given geometry and forcings, the main adjustable parameters controlling the dynamics are the plume width $W_p$, the shelf exchange parameter $C_0$, the iceberg melt parameter $M_0$ and the vertical mixing parameter $K_0$.

## 4 Validation against MITgcm

We provide an initial validation of FjordRPM by comparing it with the established general circulation model MITgcm in both idealised (section 4.1) and realistic (section 4.2) cases. In the idealised cases we consider a simple fjord-shelf geometry and simulate three fundamental modes of glacial fjord circulation: (i) buoyancy-driven circulation resulting from the input of subglacial discharge; (ii) intermediary circulation driven by shelf variability; and (iii) iceberg melt-driven circulation. In the realistic case we consider a nearly 3-year simulation of Sermilik Fjord, SE Greenland (Sanchez et al., 2024). While MITgcm is not 'the truth', it is a full three-dimensional hydrodynamic model, its dynamics are much more comprehensive than FjordRPM and it has been widely used in fjord studies (e.g., Xu et al., 2012; Sciascia et al., 2013; Carroll et al., 2015; Cowton et al., 2016; Slater et al., 2018; Zhao et al., 2022; Hager et al., 2022; Sanchez et al., 2024). It therefore provides a benchmark against which to test FjordRPM. We have allowed ourselves vary a single FjordRPM parameter to improve the comparison to the MITgcm simulations - this is the shelf exchange parameter, which takes the same value $C_0 = 1 \times 10^5$ s across all the FjordRPM simulations shown. All other parameter values used are given in Table A2.

FjordRPM has no awareness of across-fjord variability and the only awareness of along-fjord variability is that we have an estimate of horizontal velocity at the glacier (section 2.2) and fjord mouth (section 2.3). Thus the main quantities to compare





with FjordRPM are the along and across fjord-averaged temperature and salinity from MITgcm, and the fjord-shelf fluxes at
the fjord mouth from MITgcm.

## 4.1 Idealised simulations

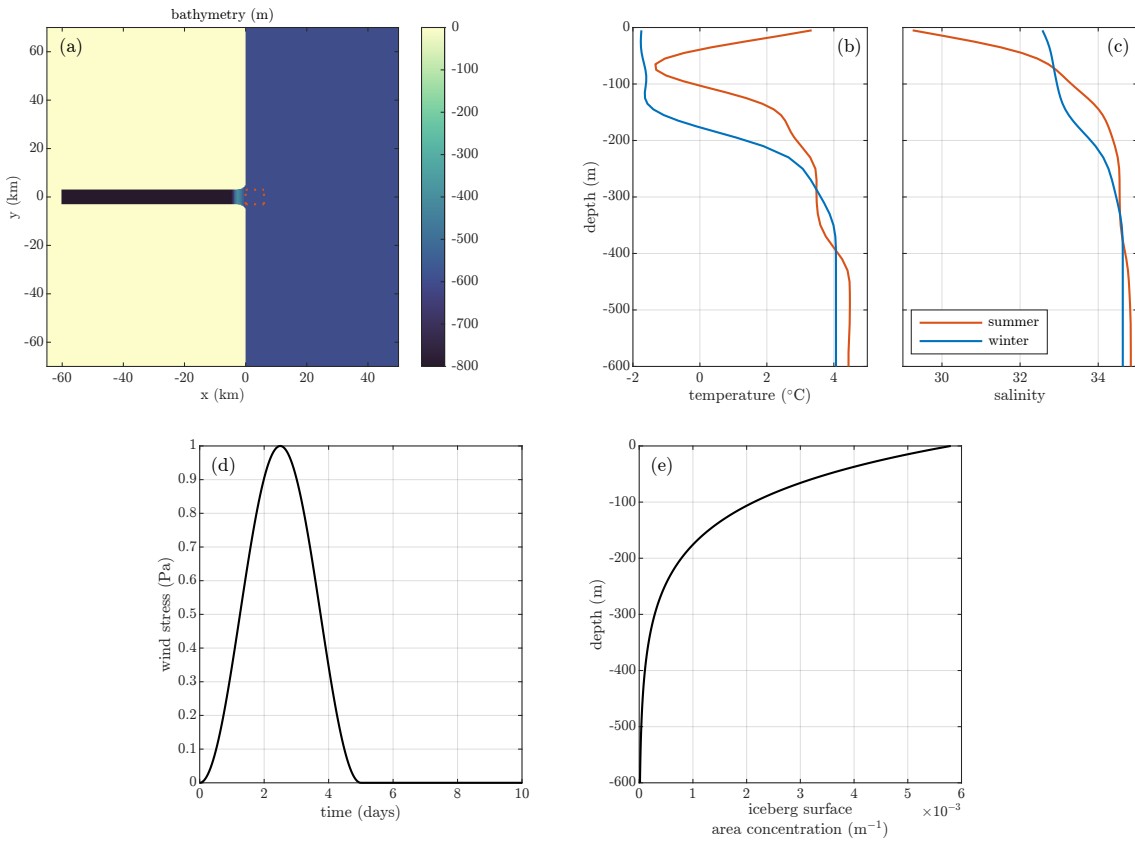

**Figure 3.** Set up of the idealised MITgcm simulations. (a) Bathymetry of part of the model domain. The red dashed square shows the region
from which we take the shelf boundary conditions for FjordRPM. (b) Temperature and (c) salinity to which the MITgcm simulations are
restored at the north, south and east boundaries, for summer and winter cases. (d) Wind stress applied to the shelf region in the intermediary
circulation simulations. (e) Iceberg ice-ocean surface area concentration profile used in the iceberg simulations.

The idealised MITgcm simulations (Fig. 3) consider a fjord that is 60 km long and uniformly 6 km wide and 800 m deep.
At the fjord head there is a single glacier with grounding line depth 800 m and at the fjord mouth there is a sill of depth 400 m.
The fjord opens out onto a shelf that is uniformly 600 m deep, 182 km wide (i.e., in the along-fjord direction) and 225 km long
(i.e., in the across-fjord direction); note that only part of the shelf is shown in Fig. 3a. The horizontal resolution is 250 m within
the fjord, telescoping to around 3 km at the far reaches of the shelf. The vertical resolution is 10 m near the surface, increasing
to 50 m at the bottom. Temperature and salinity at the boundaries of the shelf (Figs. 3b, c) are, depending on the simulation,





restored to either 'summer' or 'winter' conditions; with summer conditions coming from a conductivity-temperature-depth (CTD) profile on the continental shelf close to Sermilik Fjord, south-east Greenland and winter conditions coming from an expendable CTD in the mouth of the same fjord (Straneo et al., 2011). The initial conditions throughout the domain are taken to be the same as these far-field boundary conditions, but the simulations are run for long enough that there should be no memory of the initial conditions. The intermediary circulation simulations involve a time-varying wind stress applied to the

shelf (Fig. 3d; further details in section 4.1.2) and the iceberg melt-driven simulations require an iceberg concentration (Fig. 3e; further details in section 4.1.3). The simulations take place on an $f$-plane with $f = 1.35 \times 10^{-4}$ s$^{-1}$. In sum, these simulations are intended to be representative of large glacial fjords in Greenland.

The equivalent FjordRPM simulations use the same fjord geometry, with 60 layers of 13.3 m thickness and a time step of 0.1 days. The 'shelf' profiles $T^s(z,t)$ and $S^s(z,t)$ used to force FjordRPM are extracted from the MITgcm simulations in a

6 km × 6 km region on the shelf adjacent to the fjord (Fig. 3a). This is because shelf dynamics mean that conditions in this region can differ significantly from the MITgcm boundary conditions (Fig. 3b-c), particularly for the intermediary circulation simulations, and we wish FjordRPM to experience the same shelf forcing as the fjord in MITgcm does.

### 4.1.1 Buoyancy-driven circulation

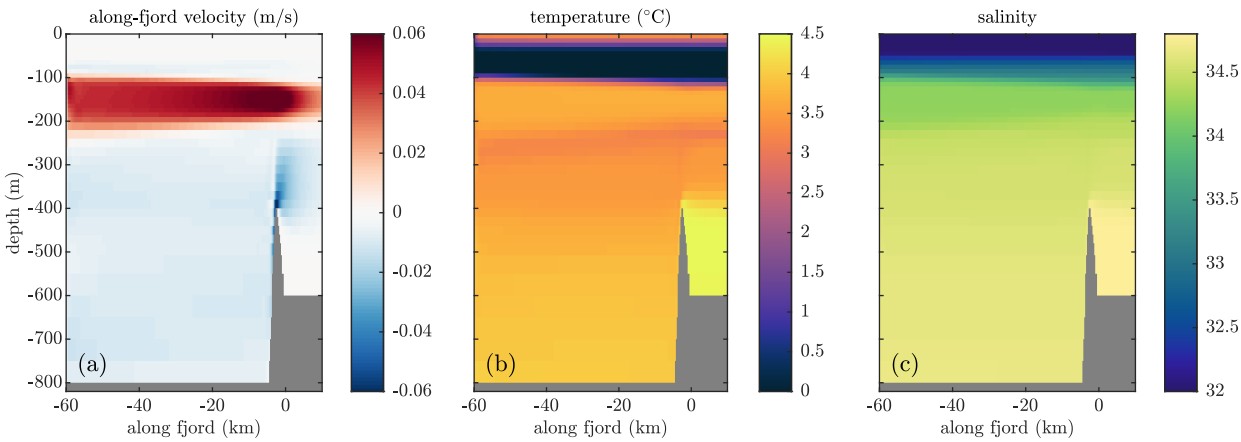

**Figure 4.** MITgcm simulation of steady state fjord dynamics obtained under sustained subglacial discharge of $Q_{sg} = 300$ m$^3$ s$^{-1}$. (a) Along-fjord velocity averaged over the width of the fjord. Positive velocities are directed from fjord to shelf. (b) Temperature and (c) salinity averaged over the width of the fjord. For $x$-axis values greater than 0 we are out on the shelf and there is no fjord width to average over, but we average over the same range of $y$-values as if we were in the fjord.

Our first test case is the buoyancy-driven circulation driven by the input of subglacial discharge from beneath tidewater

glaciers, thought to be dominant in Greenland's glacial fjords during summer. In MITgcm, the input of subglacial discharge and the resulting plume are represented using the 'Iceplume' package (Cowton et al., 2015), and the boundary conditions at the edge of the MITgcm domain are the summer conditions (Fig. 3b, c). In both MITgcm and FjordRPM, the plume width is





$W_p = 500$ m and we consider three values of subglacial discharge: $Q_{sg} = 100$, 300 or 900 m$^3$ s$^{-1}$. The subglacial discharge is held constant in time for the full 400 day length of the simulations. This is clearly longer than a melt season but it enables the
simulations to reach a steady state, which provides a good state for comparison. There is no wind stress over the shelf and no icebergs in the fjord. The plots use properties averaged over days 390-400 of the simulations.

The result obtained using MITgcm in the $Q_{sg} = 300$ m$^3$ s$^{-1}$ case is shown in Fig. 4. The dynamics are as expected from previous studies, with up-fjord flow at depth due to entrainment into the plume and down-fjord flow of plume waters closer to the surface, with an intensification of these currents over the sill. In temperature and salinity, the sill blocks the deep warm
and salty water on the shelf from getting into the fjord. Properties above the sill are relatively similar between fjord and shelf, though upwelling by the plume results in a slight warm and salty anomaly in the out-flowing layer.

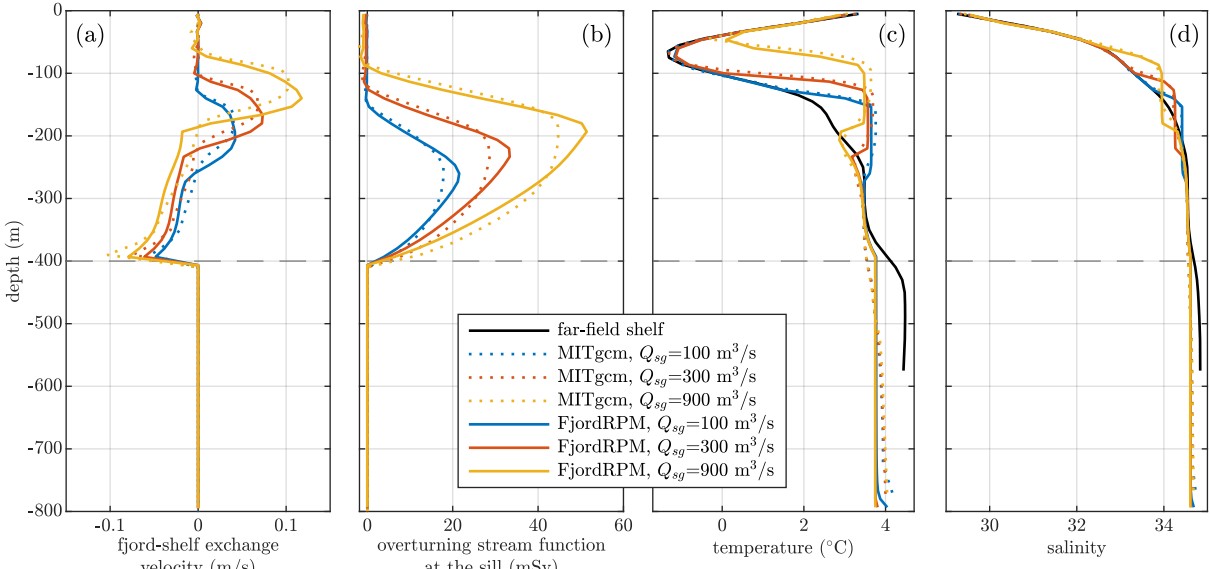

**Figure 5.** Comparison of MITgcm and FjordRPM when simulating subglacial discharge buoyancy-driven circulation. (a) Fjord-shelf exchange velocity (i.e., along-fjord velocity at the sill, averaged across the width of the fjord). Positive velocities are directed from fjord to shelf. (b) Overturning stream function at the sill. The small non-zero residual at the surface is the net flux due to the input of subglacial discharge into the fjord. (c) Temperature and (d) salinity, averaged over the length and width of the fjord. The grey dashed horizontal line in all panels denotes the sill depth.

A comparison of MITgcm and FjordRPM is shown in Fig. 5. Simulations for all three values of subglacial discharge are shown, but are qualitatively similar. FjordRPM captures the profile of fjord-shelf exchange velocity over the sill very well, including the inflow at depth, the shallower outflow and the negligible exchange close to the surface (Fig. 5a). Even details
in MITgcm, such as the roughly gaussian velocity profile of the outflow and the intensification of inflow just above the sill are present in FjordRPM. The match in fjord-average temperature and salinity is also very good (Fig. 5c & d); in particular, FjordRPM captures the divergence in fjord-to-shelf properties below sill depth, the warm and salty anomaly in the outflowing



layer and the match of fjord-to-shelf properties close to the surface. FjordRPM also skilfully captures the impact of varying subglacial discharge, mirroring the strengthening of the circulation and shallowing of the out-flowing layer under increasing

subglacial discharge.

Turning to the differences between MITgcm and FjordRPM, which are minor, we note that the outflows in FjordRPM are consistently slightly deeper than in MITgcm and the intensification of the inflow just above the sill is stronger in MITgcm. This results in a slightly different shape of the overturning stream function (Fig. 5b). Overall, however, FjordRPM captures the dynamics present in MITgcm extremely well.

### 4.1.2 Intermediary circulation

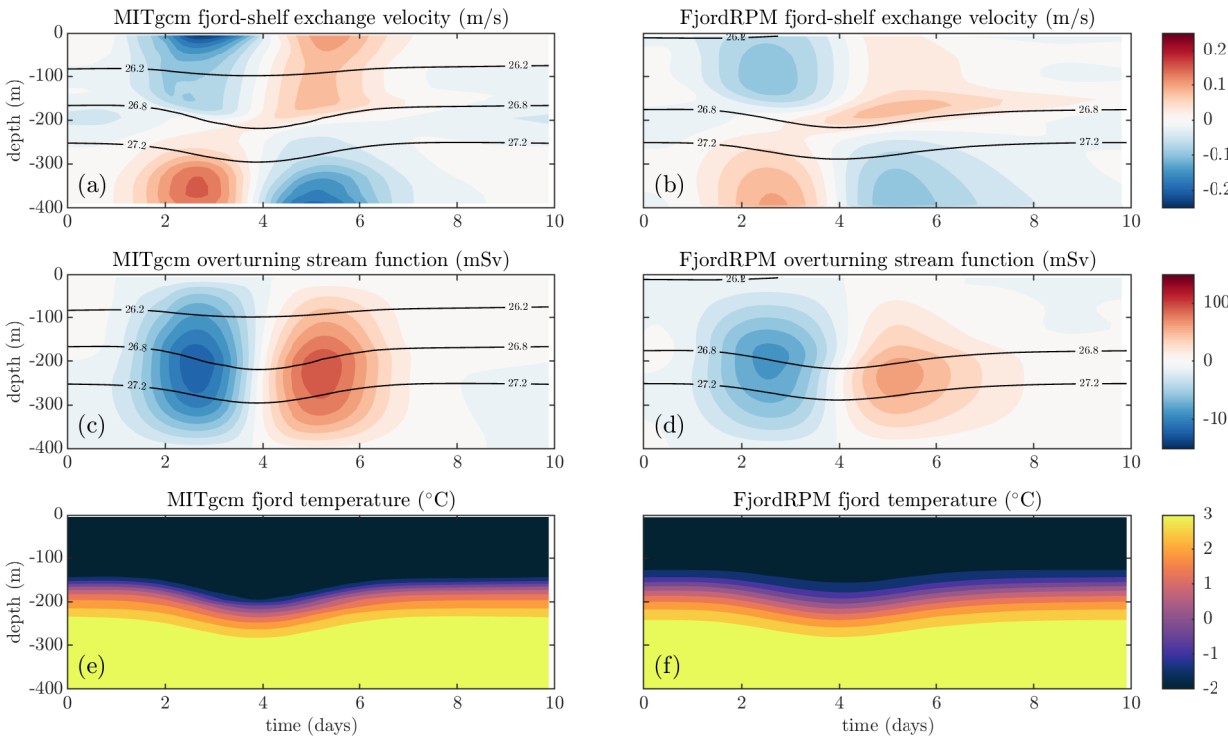

**Figure 6.** Comparison of MITgcm and FjordRPM when simulating the fjord response to wind-driven shelf variability. The left column is MITgcm results, the right column is FjordRPM results. Note that only the depth range above the sill is shown. (a) and (b) Hovmöller plots of exchange velocity over the sill, averaged across the width of the fjord. Positive velocities are directed out of the fjord and the contours show potential density. (c) and (d) equivalent plots of the overturning stream function over the sill. (e) and (f) Hovmoller plots of temperature averaged across the length and width of the fjord.

The second test case is intermediary circulation driven by variability in conditions on the shelf. The MITgcm boundary conditions are the winter profiles from Figs. 3b-c since the classic instance of intermediary circulation is driven by strong





winds over the continental shelf and this is more common in winter (e.g., Jackson et al., 2014). To produce these dynamics in MITgcm, we impose a temporally-varying southwards wind stress on the 'shelf' portion of the domain in which, with a return

time of 10 days, there is a wind event that lasts 5 days with a stress that peaks at 1 Pa after 2.5 days (Fig. 3d). After around 5 such cycles (total 50 days), the fjord in MITgcm reaches a state where the dynamics repeat themselves every 10 days and the initial conditions have been forgotten. There are no icebergs and no subglacial discharge in this simulation. We analyse the period between days 40 and 50 of the simulation.

In MITgcm, for the first 4 days of the 10-day cycle (Fig. 3d), the southward wind deepens shelf isopycnals close to the

coast, driving flow into the fjord in the upper layer and out of the fjord at depth (Fig. 6a), associated with negative values of the overturning stream function approaching 150 mSv (Fig. 6c). As the fjord adjusts to the shelf, the isopycnals in the fjord deepen and the layer of cold water at the surface thickens (Fig. 6c, e). Once the wind relaxes, after 4 days of the cycle, the circulation reverses and the fjord rebounds to the pre-wind state (Fig. 6a, c, e).

When forced by the time-varying shelf properties taken from the fjord mouth in the MITgcm simulations (red box in Fig. 3a),

FjordRPM captures the same key features of the intermediary circulation, including the direction, timing and vertical structure of the circulation (Fig. 6b, d, f). There are differences here though, with the FjordRPM fjord-shelf exchange generally more sluggish and less surface-intensified than in MITgcm (Fig. 6a vs b), giving a weaker overturning stream function (Fig. 6c vs d) and less pronounced deepening then shallowing of isopynals and isothermals (Fig. 6). The inflowing and outflowing layers are consistently separated by the $26.8\,\mathrm{kg\,m^{-3}}$ potential density contour (Fig. 6a & b); using this to separate the flow into upper

and lower layers then plotting the upper layer volume flux shows that FjordRPM captures the timing of the circulation well, but underpredicts the strength of the circulation by around 30% compared to MITgcm (Fig. 7a). As a consequence, the mean temperature of the fjord through the wind event varies less in FjordRPM than in MITgcm (Fig. 7b).

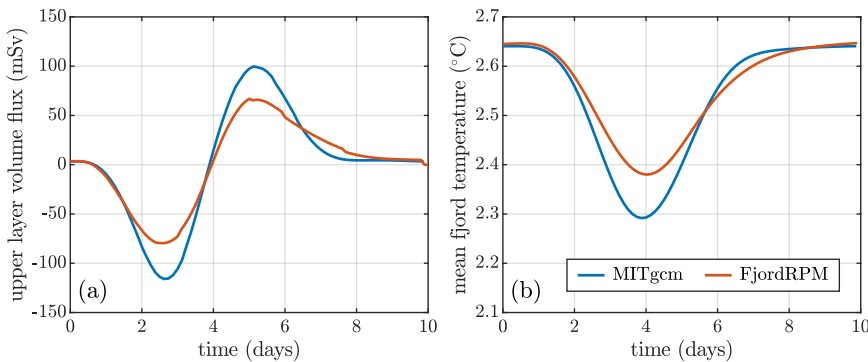

**Figure 7.** Further comparison of MITgcm and FjordRPM in simulating intermediary circulation. (a) Volume flux between the the 26.8 $\mathrm{kg\,m^{-3}}$ potential density contour (Fig. 6a, b) and the surface. Positive values are directed into the fjord. (b) Fjord temperature, averaged over fjord length, width and depth, through the 10 day wind cycle.

Improving the match between MITgcm and FjordRPM for these dynamics is not easy. The obvious parameter with which to control the fjord-shelf exchange in FjordRPM is $C_0$ (section 2.3) - in general, increasing $C_0$ will strengthen the fjord-shelf





exchange. In this case, however, we found that increasing $C_0$ does not significantly strengthen the fjord-shelf exchange. We hypothesise that this is because the exchange is, or becomes, large enough to dampen the fjord-shelf pressure gradients that drive the flow. We can therefore say that FjordRPM captures the intermediary circulation relatively well, but appears slightly sluggish compared to MITgcm, and we return to this point in the discussion.

### 4.1.3    Iceberg melt-driven circulation

Lastly, we compare the ability of MITgcm and FjordRPM to capture circulation driven by iceberg melting. In MITgcm, the icebergs are represented by the 'Iceberg' package (Davison et al., 2020), in which the icebergs are thermodynamically active but not mechanically active. That is, the simulation accounts for the cooling and freshening effect of iceberg meltwater but not for the drag exerted on the circulation by the presence of icebergs. MITgcm is configured to have the same iceberg melt parameterisation as FjordRPM (Eq. 19). The distribution of icebergs in the fjord is set by the iceberg surface area concentration

profile $\eta(z) = \frac{2 \times 10^6}{WL} \exp(-z/100)$, where $L = 60$ km is the fjord length and $W = 6$ km is the fjord width (Fig. 3e). The surface area of icebergs in an MITgcm grid cell with dimensions $(\delta x, \delta y, \delta z)$ is then $\eta(z)\delta x \, \delta y \, \delta z$, with units m$^2$. The iceberg concentration in MITgcm is set to be uniform in the horizontal. In FjordRPM, the surface area of icebergs in layer $j$ is $I_j = \eta(z_j)WLH_j$, where $z_j$ is the depth of layer $j$. The total submerged surface area of icebergs in this set-up is 200 km$^2$, consistent with observational estimates for a large Greenland fjord such as Sermilik (Enderlin et al., 2016). The simulations

are run with both the summer and winter boundary conditions from Fig. 3b, c. There is no subglacial discharge and no wind stress on the shelf.

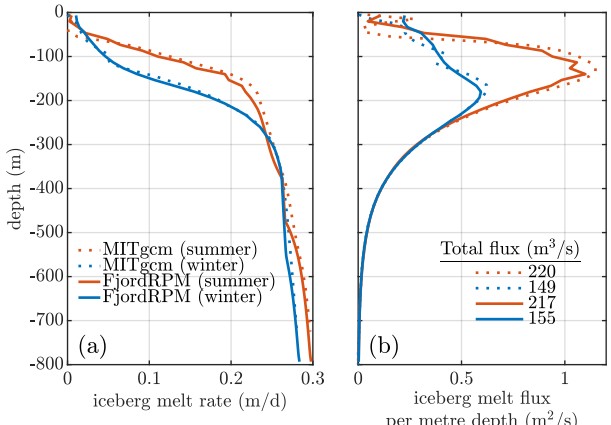

**Figure 8.** Comparison of MITgcm and FjordRPM-simulated (a) iceberg melt rate, and (b) iceberg melt flux in the simulation of fjord dynamics driven by iceberg melt.

    The obtained iceberg melt rates and fluxes in MITgcm and FjordRPM are shown in Fig. 8. The melt rates and fluxes are comparable with observations from Sermilik Fjord (Enderlin et al., 2016); this is no coincidence as it is how we set our suggested value of the iceberg melt parameter, $M_0 = 5 \times 10^{-7} \, \text{m s}^{-1} \, (^\circ\text{C})^{-1}$ (Table A2). MITgcm and FjordRPM show mostly





close agreement in both melt rate and melt flux. Given that the two methods of simulation have been configured with the same iceberg distribution and the same melt parameterisation (which depends only on thermal forcing), this is not surprising and is only really a test of the modelled fjord temperature.

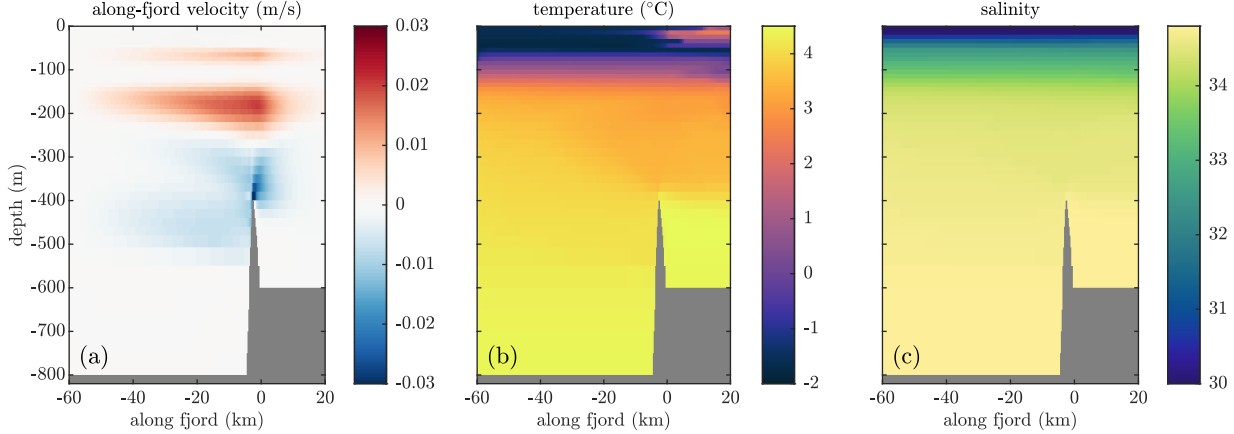

**Figure 9.** MITgcm simulation of steady-state fjord dynamics driven by iceberg melting with summer shelf boundary conditions. (a) Along-fjord velocity averaged over the width of the fjord. Positive velocities are directed from fjord to shelf. (b) Temperature and (c) salinity averaged over the width of the fjord.

  A tougher test of FjordRPM comes from comparing the dynamics induced by the melting icebergs. In MITgcm with the summer shelf boundary conditions, the dominant feature is an overturning cell at intermediate depth with an outflow centered

at 200 m (Fig. 9a & 10a-b). We interpret that this overturning cell is driven by deep iceberg melt (Fig. 8) that is able to upwell through the relatively unstratified waters at depth before ceasing to rise and flowing horizontally below the more stratified upper 200 m (Fig. 9c & 10d). The upwelling entrains ambient fjord waters, setting up the circulation cell and gives a small warm anomaly in the fjord relative to the shelf at 200 m depth (Fig. 10c). The presence of iceberg melt and upwelling below sill depth (Fig. 8) leads to some of the below-sill water being replaced, so that the fjord becomes slightly cooler and fresher

than the shelf just below sill depth (Fig. 10c-d). Finally, the summer MITgcm simulation shows a strong cooling of the fjord relative to the shelf at the surface (Fig. 10c).

  In MITgcm with the winter shelf boundary conditions (the winter equivalent of Fig. 9 is not shown), the cooler surface waters mean that iceberg melt rates are lower than in summer (Fig. 8). The overturning stream function has a second maximum located at 80 m depth (Fig. 10b) that we interpret is driven by iceberg upwelling in the less-stratified near-surface waters that

are present in winter (Fig. 10d).

  Overall, FjordRPM does very well at representing both the summer and winter dynamics (Fig. 10), capturing (i) the main and secondary circulation cells, (ii) the strength of these cells, (iii) the warm anomalies in the fjord at intermediate depths, (iv) the cold anomaly in the fjord at the surface in summer, and (v) the below-sill cooling and freshening. Considering how these dynamics come about, in MITgcm, cells with icebergs are freshened and cooled and any upwelling results from a solution of



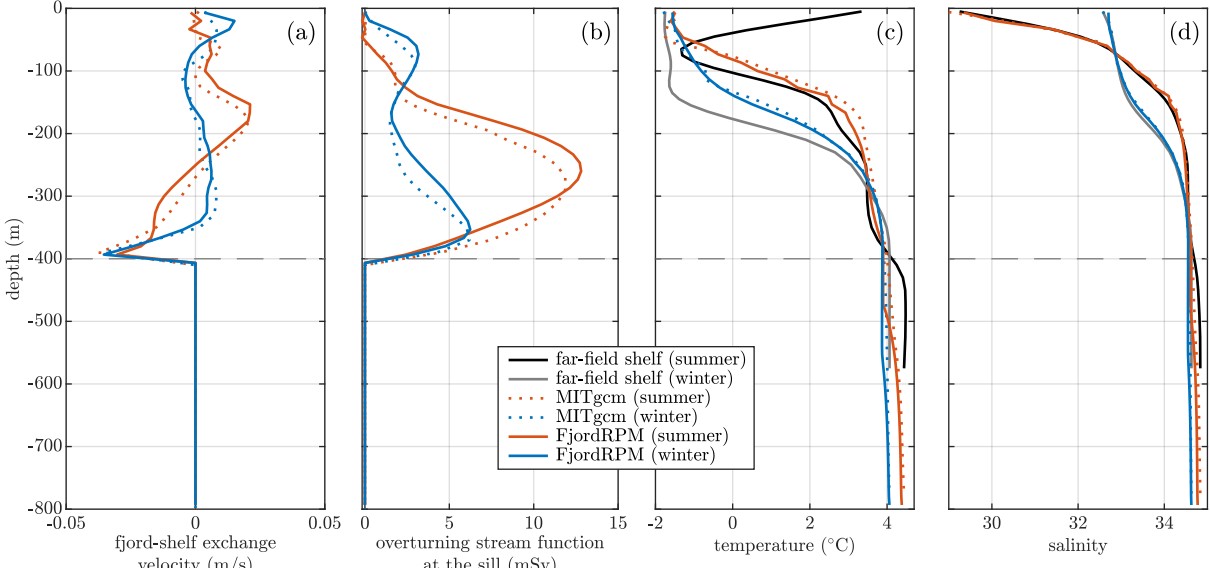

**Figure 10.** Comparison of MITgcm and FjordRPM when simulating circulation driven by iceberg melting. (a) Fjord-shelf exchange velocity. (b) Overturning stream function at the sill. (c) Temperature and (d) salinity, averaged over the length and width of the fjord. The grey dashed horizontal line in all panels denotes the sill depth.

the full hydrodynamic equations (note however that these plumes of upwelling iceberg meltwater are not properly resolved in a fjord-scale MITgcm simulation). In FjordRPM, upwelling is simply parameterised as described in section 2.4. Given these very different approaches, it is encouraging that FjordRPM matches the MITgcm simulations so well.

## 4.2   Realistic simulation of Sermilik Fjord

As a final test of FjordRPM, we compare to a nearly 3-year, realistic MITgcm simulation of Sermilik Fjord and the adjacent
shelf in south-east Greenland (Fig. 11), conducted by Sanchez et al. (2024). In this context, "realistic" means that their simulation used the actual bathymetry of the fjord and shelf and was forced at the glacier by subglacial discharge from a regional climate model (RACMO; Noël et al., 2018) at the surface by an atmospheric reanalysis (ERA5; Hersbach et al., 2020) and at the ocean boundary by an ocean reanalysis (ASTE; Nguyen et al., 2021). Together with observational validation from this well-studied fjord, the simulations in Sanchez et al. (2024) are as close to realistic as we can currently get (aside from the
presence of icebergs, which they did not consider). Full details may be found in their paper.

We attempt to undertake the same simulation in FjordRPM, making the set-up as close as possible given the much-simplified nature of the reduced physics model. Where MITgcm used realistic bathymetry (Fig. 11), FjordRPM has no sill and comprises a cuboid fjord of width 6 km, depth 900 m and length 101 km (ensuring that the total volume of the fjord is the same in MITgcm and FjordRPM). Where in MITgcm there are three glaciers with plumes in side-arms of the main fjord, FjordRPM has three





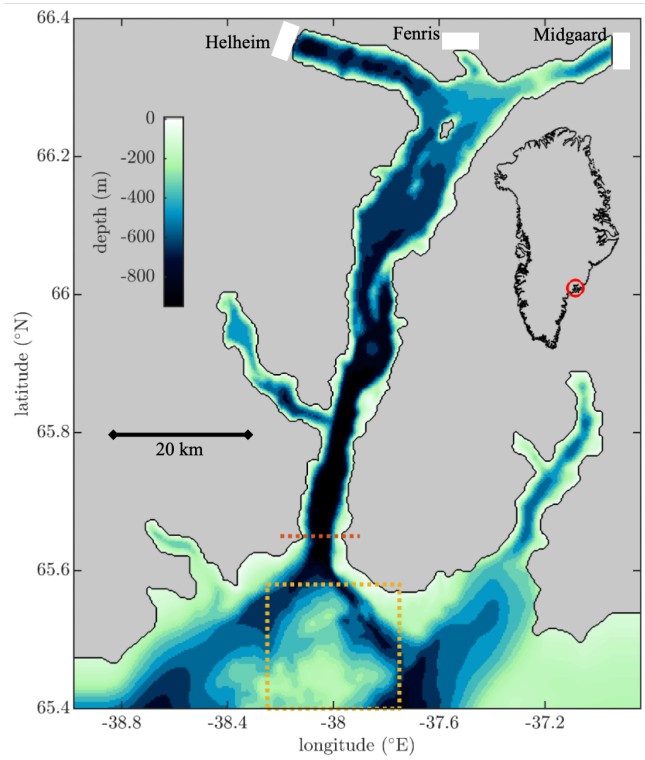

**Figure 11.** Sermilik Fjord, south-east Greenland, as modeled by Sanchez et al. (2024). Note that their model domain included a larger region of the shelf than is indicated by this plot. The white annotated patches show the location of the three main glaciers that discharge into the fjord. The red dotted line shows the fjord mouth flux gate and the yellow dashed square shows the region from which we extract properties from MITgcm to force FjordRPM. The inset shows the location in south-east Greenland.

plumes in the cuboid fjord with respective initiation depths of 650, 160 and 465 m, corresponding to the grounding line depths of the three glaciers. The plume width is the same in MITgcm and FjordRPM, at 280 m, and the subglacial discharge forcing is identical. Unlike MITgcm, FjordRPM has no surface forcing. For the shelf forcing in FjordRPM, we extract properties from the MITgcm simulation close to the fjord mouth (Fig. 11). We configure FjordRPM with 60 layers of 15 m thickness, a time step of 0.1 days and with parameters as in Table A2. The FjordRPM simulation runs in a few seconds on a laptop.

We compare MITgcm and FjordRPM in terms of simulated fjord properties (Fig. 12) and fjord-shelf exchange fluxes (Fig. 13). The fjord properties extracted from MITgcm are averaged over all model points up-fjord of a flux gate at the fjord mouth (Fig. 11), while from FjordRPM they are simply the layer properties. The exchange fluxes from MITgcm are those passing through the flux gate at the fjord mouth, while from FjordRPM they are the fjord-shelf exchange fluxes.

When compared to MITgcm, FjordRPM captures the evolution of fjord properties relatively well (Fig. 12). It captures the
timing and magnitude of the seasonal cycle in both temperature and salinity at all depths. It also recreates the high-frequency variability in properties that arises during winter storm systems (e.g., Fig. 12e). There are differences between MITgcm and





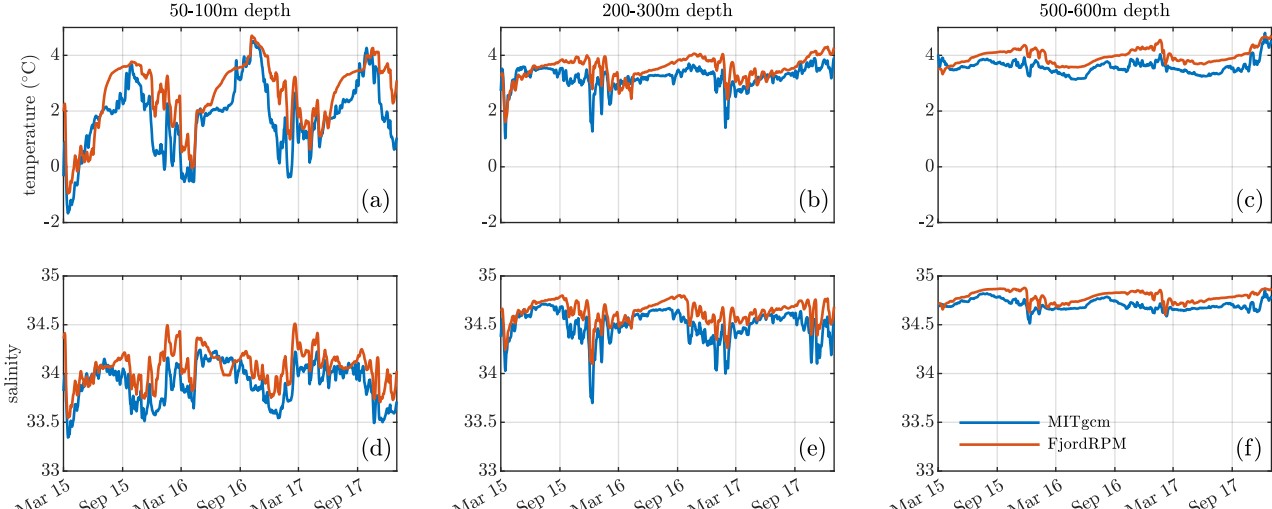

**Figure 12.** Sermilik Fjord properties from March 2015 to December 2017, as modeled by MITgcm (blue, Sanchez et al., 2024) and by FjordRPM when forced by shelf properties from MITgcm (red). (a)-(c) temperature averaged over the indicated depths; (d)-(f) salinity averaged over the same depths.

FjordRPM – in general, FjordRPM appears warmer and saltier than MITgcm. There are also specific depths and times where the properties diverge, for example during summer at 50–100 m depth (Fig. 12a), where FjordRPM becomes warmer than MITgcm, likely due to differences in plume neutral buoyancy. We have not dug further into these differences, as to do so

properly would require more space that feels appropriate for the present manuscript, but possible reasons for the differences include the specifics of how the shelf forcing is sampled (Fig. 11) and the lack of realistic hypsometry in FjordRPM.

FjordRPM also does well at capturing fjord to shelf exchange at approximately monthly or longer timescales, when compared to MITgcm (Fig. 13, thicker lines). FjordRPM captures the seasonal cycle in fjord to shelf volume exchange (Fig. 13a), with increased exchange during winter and more quiescent periods during summer. Considering freshwater fluxes, FjordRPM

captures the variable exchange of freshwater during winter and the emergence of a net freshwater flux of 1-5 mSv from the fjord to the shelf during summer (Fig. 13b). In fjord to shelf heat exchange, FjordRPM has much of the same variability as MITgcm, but in terms of absolute value, FjordRPM exceeds MITgcm during summer and MITgcm exceeds FjordRPM during winter (Fig. 13c). We suspect these differences may result from air-sea heat fluxes, which are included in MITgcm but not in FjordRPM – indeed, excluding the surface 30 m from the exchange flux changes the sign of the heat flux in MITgcm during

summer, leading to better agreement between the models (Fig. 13d).

The greatest differences in exchange fluxes between FjordRPM and MITgcm are at weekly or shorter timescales (Fig. 13, thinner lines), where MITgcm shows a greater magnitude of variability, especially during summer. We have forced FjordRPM with daily output from the MITgcm simulation – it is possible that FjordRPM would show greater variability at short timescales if we extracted, say, hourly output from MITgcm. But, this 'sluggish' nature of FjordRPM appears similar to the idealised



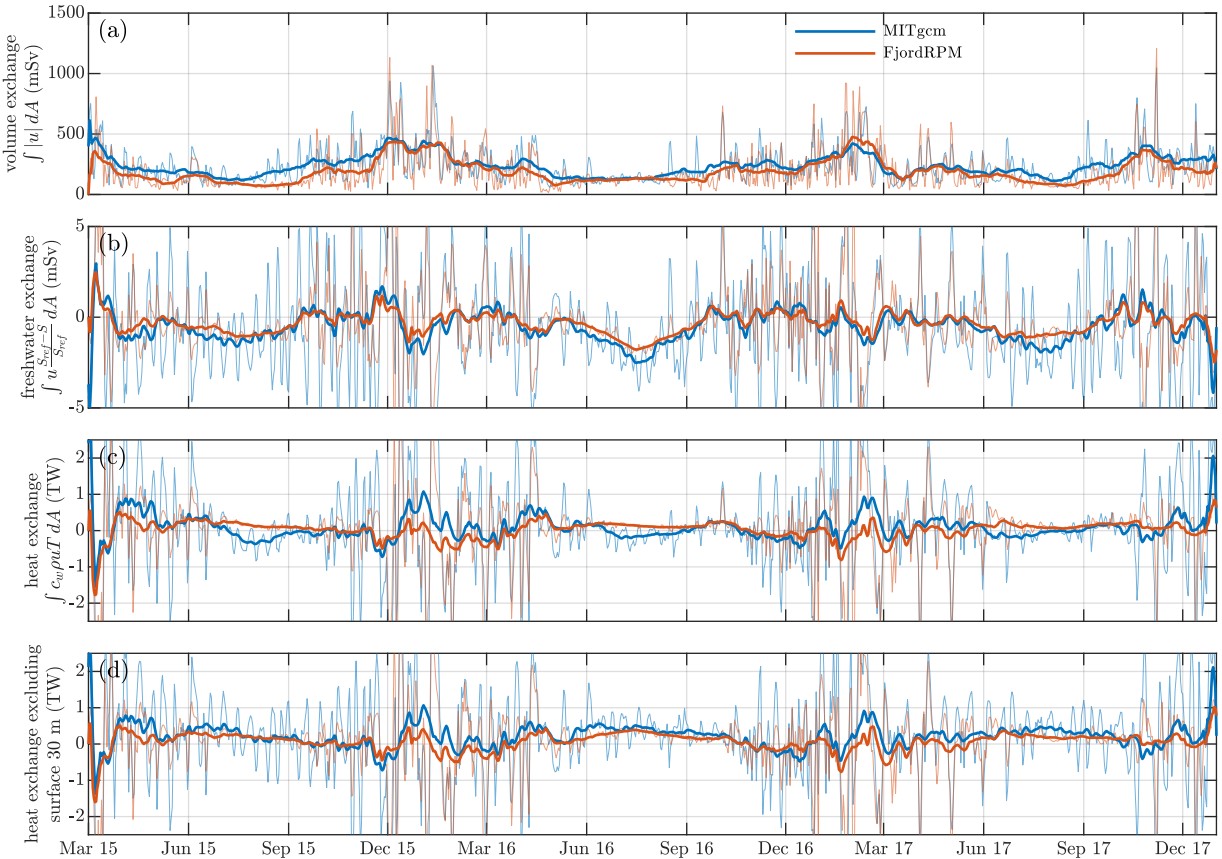

**Figure 13.** Fluxes at the fjord mouth (section indicated in Fig. 11) as modeled by MITgcm (blue, Sanchez et al., 2024) and by FjordRPM when forced by shelf properties from MITgcm (red). (a) Volume exchange obtained by integrating the absolute value of the meridional velocity over the fjord mouth cross-section (i.e., integrating over the fjord width and depth). (b) Freshwater exchange assuming $S_{ref} = 34.9$, which is a typical value at the bottom of the fjord. Positive values indicate a net flux of freshwater from the shelf into the fjord. (c) Heat exchange. Positive values indicate a net flux of heat from the shelf into the fjord. (d) As for panel (c), but excluding the surface 30 m from the integration. In all plots the thin lines are daily values while the thick lines are the result of smoothing the daily values with a 30-day centered moving window.

intermediary circulation simulations above and so may represent something more fundamental about the implemented physics. We expand on this in the discussion and conclude at this point that FjordRPM shows good promise at capturing fjord properties and longer-term fjord-shelf exchange fluxes when applied to real fjords.



## 5   Discussion

Having motivated, designed, coded and tested FjordRPM, we now discuss its strengths and weaknesses together with potential
future work and applications. One aspect that has not been discussed so far is the efficiency of the model. Consisting of around
400 lines of code, the model takes around a second to run a fjord for one model year on a laptop. FjordRPM is clearly very
different in character to a general circulation model and is not suited to all purposes, but this compares to hundreds or thousands
of CPU hours to run equivalent simulations (such as the validation experiments in this manuscript) using a general circulation
model. The principal factors determining the FjordRPM run time are the number of layers (i.e., the vertical resolution), the
time step and how frequently the subglacial discharge plume dynamics are updated.

When compared to several idealised and a single realistic MITgcm simulation, FjordRPM shows excellent ability to capture
fjord properties and circulation. Specifically, FjordRPM shows ability to capture (i) the influence of a sill on below-sill fjord
temperature and salinity (Fig. 5); (ii) the influence of subglacial discharge-driven upwelling on fjord properties in the upper
water column (Fig. 5); (iii) the influence of synoptic (Figs. 6, 7 & 12) and seasonal (Fig. 12) shelf variability on fjord properties;
and (iv) the role of icebergs in cooling, freshening and upwelling of fjord properties (Fig. 10). Furthermore, FjordRPM captures
very well the exchange between fjord and shelf – in both a depth-resolved and depth-integrated manner – induced by (i)
subglacial discharge (Fig. 5), (ii) shelf variability (Figs. 6, 7 & 13) and (iii) iceberg melt (Fig. 10). The listed dynamics
comprise many of the processes understood to be important in glacial fjords.

There are also aspects in the comparison to MITgcm in which FjordRPM is less successful. FjordRPM is generally sluggish
compared to MITgcm in its response to synoptic shelf variability, as seen in both the idealised intermediary circulation simu-
lations (Figs. 6 & 7) and the realistic simulation of Sermilik Fjord (Fig. 13). From the implemented physics, we might expect
FjordRPM to be more sluggish in this manner – the fjord-shelf exchange fluxes (section 2.3) are based on a steady-state balance
that does not consider an acceleration term. In addition, FjordRPM has no along-fjord variability, so waters that come into the
fjord from the shelf are instantly mixed along the full fjord, limiting the fjord-shelf density gradient. In contrast, MITgcm has
along-fjord resolution, which may allow it to have high fjord-shelf density gradients purely in the region of the fjord mouth,
leading to higher frequency and magnitude fjord-shelf exchange.

A related issue is how exactly to derive the shelf boundary conditions required by FjordRPM. In the validation against
MITgcm, we have sampled these boundary conditions from a region on the shelf very close to the fjord mouth (Figs. 3 & 11).
However, there can be spatial variability in water properties, particularly in the presence of variable bathymetry (e.g., Fig. 11).
We found that adjusting the definition of the shelf sampling region, say by making the region smaller or larger or moving it
around, changes the FjordRPM results quantitatively but not qualitatively. That is, the plume neutral buoyancy level may adjust,
the fjord-shelf exchange fluxes may increase or decrease, and the fjord water properties may warm or cool, but the magnitude
of these changes is small compared to the main features and variability in the FjordRPM solution.

A broader point, however, is whether it is necessary to have a well-resolved simulation of the continental shelf to provide the
shelf boundary conditions for FjordRPM. The comparisons presented in this paper have shown very good agreement between
FjordRPM and MITgcm in the fjord when FjordRPM is forced with water properties extracted from MITgcm just outside the




fjord mouth, but as illustrated in the MITgcm simulation of Sermilik Fjord (Sanchez et al 2024) these fjord mouth properties may themselves be substantially modified during cross-shelf transit in ways which are not effectively captured by relatively coarse ocean reanalyses or projections. Future work could seek to quantify to what extent this is a significant issue; that is,

what difference it makes to the FjordRPM solution when we force it with a coarse versus resolved representation of the shelf outside the fjord. If there is a significant difference, a further link in the model chain may then be required, either using higher resolution regional simulations (Verjans et al., 2023) or simplified models of shelf processes more akin to the philosophy of FjordRPM.

There are similarly many avenues for further development of the model. One obvious aspect is to implement realistic fjord

hypsometry. At present, the fjord is assumed to be a cuboid that we can adjust to ensure it has the same overall volume as a real-world fjord (e.g., section 4.2), but we cannot ensure it has the right volume at all depths. Another prominent aspect for improvement is the fjord-shelf exchange parameterisation (section 2.3). Previous work has suggested that depending on the geometry, stratification and circulation, fjord-shelf exchange can be dictated by geostrophic balance, hydraulic control or friction and mixing (Zhao et al., 2021; Nilsson et al., 2023; Sanchez et al., 2023). We have found that FjordRPM appears

very effective at capturing a range of circulations with a very simple fjord-shelf exchange parameterisation (section 2.3), but its somewhat empirical nature means that it may need tuned individually for each fjord. A more sophisticated fjord-shelf exchange parameterisation, and a further set of validation experiments, could seek to capture various fjord-shelf dynamical balances in one framework and allow a 'universal' tuning of this exchange parameterisation.

There are also fjord processes that we have not represented at all, but which may be important in particular systems or

for particular research questions. These include ambient (outside-of-plume) melting of the glacier front, fluxes of heat and freshwater at the fjord surface, sea ice, wind stress and tides. Iceberg concentration, currently prescribed by a static ice-ocean surface area, could be made to evolve on the balance of input from calving, loss to melting and export to the shelf.

Another avenue for future work concerns validation of the model beyond the initial tests that we have undertaken here. This could take the form of further MITgcm simulations that are targeted to test specific aspects of the exchange parameterisations,

but it would be particularly valuable to test FjordRPM against observations, either looking at the model's ability to capture fjord-to-fjord differences (say using CTD profiles from many of Greenland's fjords), or its ability to capture variability in depth and time at an individual fjord (say using mooring data).

Lastly, we return to the potential applications of FjordRPM. While being mindful of the issues just discussed, the ability to capture fjord temperature under a range of forcings (Figs. 5, 7, 10 & 12) means it could be used to generate ocean boundary

conditions for ice sheet models. Similarly, the ability to capture freshwater export to the shelf (Figs. 5, 7, 10 & 13) suggests it could be used to generate freshwater boundary conditions for ocean models. In both directions, it offers a means of coupling ice sheet and ocean models in a practical and efficient manner. Due to the simplicity and transparent nature of the dynamics (e.g., the ability to easily turn on or off, or tweak, individual processes), FjordRPM may also have a role in investigating fjord processes.





# 6    Conclusion

We have presented the design and initial validation of FjordRPM – a reduced physics model for glacial fjords. The model consists of a number of vertically stacked layers, each of which extends over the full width and length of the fjord at a given depth. Exchanges of volume, heat and salt into and out of layers are parametrised to represent important fjord processes, including subglacial discharge-driven upwelling and submarine melting, exchange with the shelf, iceberg-driven upwelling and

melting, vertical mixing and vertical advection. The model thus lies somewhere between a box model and a "1.5-dimensional" model (i.e., a 1-dimensional model with lateral exchanges). The model has the capability to represent a sill and multiple glaciers in a single fjord.

We have validated FjordRPM by comparing to idealised and realistic simulations in the established general circulation model MITgcm, finding that FjordRPM is successful in capturing almost all tested aspects of glacial fjord properties and circulation,

but is sluggish in response to high-frequency shelf variability. FjordRPM is also highly efficient, taking around a second per fjord per model year on a laptop, in contrast to the hundreds or thousands of CPU hours required for MITgcm to run the equivalent simulations. While further development and validation would be valuable, we present FjordRPM as a promising tool for conceptualising fjord dynamics, and for linking ice masses and the ocean in ice and climate projection efforts.





## Appendix A: Scalings for iceberg-driven upwelling

Within an unstratified body of water (e.g., within each FjordRPM layer), Magorrian and Wells (2016) show in their Eq. 31 that up to a constant, the upwelling velocity scales as

$$u \sim \left( \frac{\Delta \rho_u}{\rho_I} g \sin \phi \, X \right)^{1/2} \tag{A1}$$

in which $\Delta \rho_u$ is the density difference between the upwelling plume and the ambient, $\rho_I$ is a reference density, $g$ is gravity, $\phi$ is the angle of the interface measured from the horizontal and $X$ is the along-ice distance. To derive a simple scaling for

FjordRPM we take $X$ to be the layer thickness $H_j$ and assume the iceberg sides are vertical so that $\sin \phi = 1$. Continuing to follow Magorrian and Wells (2016), the density difference is given by

$$\Delta \rho_u = \frac{M_u}{\alpha_i + M_u} \Delta \rho_{I0}^{ef} \tag{A2}$$

where $\alpha_i$ is the entrainment coefficient as in section 2.4, $M_u = \dot{m}/u$ is the constant that relates velocity and melt rate and $\Delta \rho_{I0}^{ef}$ is the density difference between meltwater and the ambient. Assuming $M_u \ll \alpha_i$ and substituting Eq. A2 into Eq. A1, we get

$$u \sim \left( \frac{g \Delta \rho_{I0}^{ef}}{\rho_I} \frac{\dot{m}}{\alpha_i u} H_j \right)^{1/2} \tag{A3}$$

Now we note that $g \Delta \rho_{I0}^{ef}/\rho_I$ is $g'_{j,melt}$ as given by Eq. 21 and that $\dot{m}$ can be written as $Q_j^{melt}/I_j$ (i.e., the melt rate is the melt flux divided by the surface area undergoing melting). We can then solve for $u$ to obtain the estimate of the upwelling velocity that is used in the model (Eq. 20)

$$u \equiv v_j \sim \left[ \frac{Q_j^{melt} g'_{j,melt} H_j}{\alpha_i I_j} \right]^{1/3} \tag{A4}$$

To obtain the length scale that controls how far the upwelling rises, start from Magorrian and Wells (2016) Eq. 37

$$l_\rho = \Delta \rho_u \left( \frac{\partial \rho_a}{\partial X} \right)^{-1} \tag{A5}$$

Using the same substitutions as for the upwelling velocity, this may be written as (again under $M_u \ll \alpha_i$)

$$l_\rho = \frac{M_u}{\alpha_i + M_u} \Delta \rho_{I0}^{ef} \left( \frac{\partial \rho_a}{\partial X} \right)^{-1} = \frac{\dot{m}}{\alpha_i u} \frac{g'_{j,melt} \rho_I}{g} \left( \frac{\partial \rho_a}{\partial X} \right)^{-1} = \frac{Q_j^{melt} g'_{j,melt}}{\alpha_i I_j u} \left( \frac{\partial g'}{\partial X} \right)^{-1} \tag{A6}$$

where in the last step we have used the fact that $g' = g \, \delta \rho / \rho_{ref}$ (Eq. 1). We can then use Eq. A4 to rewrite this as

$$l_\rho = \frac{v_j^2}{H_j} \left( \frac{\partial g'}{\partial X} \right)^{-1} \tag{A7}$$

If we finally estimate the derivative for layer $j$ using the first-order finite difference

$$\frac{\partial g'}{\partial X} \approx \frac{g'_{j+1,j}}{\frac{1}{2} (H_j + H_{j+1})} \tag{A8}$$

then we get the final estimate of the upwelling length scale that is used in the model (Eq. 23)

$$l_{j+1,j}^{ice} = \frac{v_{j+1}^2}{H_{j+1}} \frac{H_j + H_{j+1}}{2 g'_{j+1,j}} \tag{A9}$$

| Symbol | Units | Description |
|---|---|---|
| $V_j$ | $m^3$ | layer volume |
| $H_j$ | m | layer thickness (∗) |
| $T_j$ & $S_j$ | °C & $g\,kg^{-1}$ | layer temperature & salinity (∗) |
| $I_j$ | $m^2$ | layer iceberg surface area (∗) |
| $Q_j^{X_y}$ | $m^3\,s^{-1}$ or $m^3\,s^{-1}$°C or $m^3\,s^{-1}\,g\,kg^{-1}$ | layer flux; $X$=type, $y$=process (∗) |
| $Q_{j+1,j}^{X_y}$ | " | layer-to-layer flux; $X$=type, $y$=process |
| $b_p$ | m | discharge plume width |
| $u_p$ | $m\,s^{-1}$ | discharge plume velocity |
| $m_p$ | $m\,s^{-1}$ | discharge plume submarine melt rate (∗) |
| $T_p$ & $S_p$ | °C & $g\,kg^{-1}$ | discharge plume temperature & salinity |
| $g'_p$ | $m\,s^{-2}$ | plume-fjord relative buoyancy |
| $T_b$ & $S_b$ | °C & $g\,kg^{-1}$ | ice-ocean boundary layer temperature & salinity |
| $Q_{sg}$ | $m^3\,s^{-1}$ | subglacial discharge volume flux (∗) |
| $Q_{sm}$ | $m^3\,s^{-1}$ | discharge plume submarine melt volume flux (∗) |
| $T^s$ & $S^s$ | °C & $g\,kg^{-1}$ | shelf temperature & salinity profile (∗) |
| $T_j^s$ & $S_j^s$ | °C & $g\,kg^{-1}$ | shelf properties averaged over depth of layer $j$ |
| $g'_{sj}$ | $m\,s^{-2}$ | shelf-fjord relative buoyancy |
| $\phi_j$ | $m^2\,s^{-2}$ | fjord-shelf potential |
| $u_b$ | $m\,s^{-1}$ | fjord-shelf velocity correction ensuring conservation of fjord volume |
| $T_j^f$ | °C | layer in-situ freezing point |
| $Q_j^{melt}$ | $m^3\,s^{-1}$ | layer iceberg melt flux (∗) |
| $v_j$ | $m\,s^{-1}$ | upwelling velocity driven by iceberg melt |
| $g'_{j,melt}$ | $m\,s^{-2}$ | meltwater-fjord relative buoyancy |
| $T_{eff}$ | °C | meltwater effective temperature |
| $Q_j^{ent}$ | $m^3\,s^{-1}$ | entrainment flux due to iceberg upwelling |
| $l_{j+1,j}^{ice}$ | m | iceberg upwelling length scale |
| $f_{j+1,j}^{ice}$ | - | iceberg upwelling fraction |
| $u_j$ | $m\,s^{-1}$ | layer horizontal velocity scale |
| Ri | - | layer-to-layer Richardson number |
| $K_z$ | $m^2\,s^{-1}$ | vertical tracer mixing diffusivity |
| $Q_j^{imb}$ | $m^3\,s^{-1}$ | layer volume imbalance due to plume, shelf and icebergs |

**Table A1.** List of FjordRPM model variables. The first section contains basic variables, the second section contains plume flux variables, the third section is shelf flux variables, the fourth section is iceberg variables, the fifth section is vertical mixing variables and the last section concerns conservation of layer volume. Starred variables are key inputs or outputs; all others are essentially working variables.





| Symbol | Units | Description | Suggested value | Source/reason |
|---|---|---|---|---|
| $g$ | $\mathrm{m\,s^{-2}}$ | gravitational acceleration | 9.81 | standard |
| $\beta_S$ | $\mathrm{(g\,kg^{-1})^{-1}}$ | haline contraction coefficient | $7.86 \times 10^{-4}$ | Jenkins (2011) |
| $\beta_T$ | $\mathrm{(^{\circ}C)^{-1}}$ | thermal expansion coefficient | $3.87 \times 10^{-5}$ | " |
| $l$ | $\mathrm{J\,kg^{-1}}$ | latent heat | $3.35 \times 10^{5}$ | " |
| $c_w$ | $\mathrm{J\,kg^{-1}(^{\circ}C)^{-1}}$ | heat capacity of seawater | 3974 | " |
| $c_i$ | $\mathrm{J\,kg^{-1}(^{\circ}C)^{-1}}$ | heat capacity of ice | 2009 | " |
| $\lambda_1$ | $\mathrm{^{\circ}C\,ppt^{-1}}$ | dependence of freezing point on salinity | $-5.73 \times 10^{-2}$ | " |
| $\lambda_2$ | $\mathrm{^{\circ}C}$ | freezing point offset | $8.32 \times 10^{-2}$ | " |
| $\lambda_3$ | $\mathrm{^{\circ}C\,m^{-1}}$ | dependence of freezing point on depth | $-7.61 \times 10^{-4}$ | " |
| $\Gamma_T$ | - | heat transfer coefficient | $2.2 \times 10^{-2}$ | " |
| $\Gamma_S$ | - | salt transfer coefficient | $6.2 \times 10^{-4}$ | " |
| $C_d$ | - | discharge plume-ice drag coefficient | $2.5 \times 10^{-3}$ | " |
| $T_i$ | $\mathrm{^{\circ}C}$ | ice temperature | -10 | Cowton et al. (2015) |
| $\alpha_p$ | - | discharge plume entrainment parameter | 0.1 | " |
| $\alpha_i$ | - | iceberg plume entrainment parameter | 0.1 | " |
| $\mathrm{Ri}_0$ | - | Richardson number cutoff for tracer mixing | 0.7 | Large et al. (1994) |
| $W_p$ | m | across-glacier discharge plume width | 250 | Jackson et al. (2017) |
| $C_0$ | s | shelf exchange efficiency | $1 \times 10^{5}$ | validation experiments in section 4 |
| $K_0$ | $\mathrm{m^2\,s^{-1}}$ | vertical tracer diffusivity scalar | $5 \times 10^{-3}$ | Large et al. (1994) |
| $K_b$ | $\mathrm{m^2\,s^{-1}}$ | vertical tracer diffusivity background | $1 \times 10^{-6}$ | adjusted from Large et al. (1994) following results in section 4 |
| $M_0$ | $\mathrm{m\,s^{-1}(^{\circ}C)^{-1}}$ | iceberg melt efficiency | $5 \times 10^{-7}$ | based on section 4.1.3 and Enderlin et al. (2016) |

**Table A2.** List of parameters in FjordRPM with suggested values. The first section of the table is intended to be physical constants that should rarely need changing, while the second section contains parameters that should be seen as adjustable, for example in tuning FjordRPM to MITgcm or observations.



*Code availability.*  The FjordRPM release (Slater et al., 2024) associated with this manuscript is available at https://doi.org/10.5281/zenodo.14536606.
The corresponding GitHub repository is at https://github.com/fjord-mix/fjordrpm. The release contains source code, example plotting code
and idealised example simulations. Instructions for installation and running are included in the readme file at these links.

*Author contributions.*  DAS, TC and MI conceived of the model and obtained the funding. DAS and EJ wrote the model with significant input
from MMB. NF ran the MITgcm simulations with support and input from TC. DAS undertook the comparison of FjordRPM and MITgcm.
DAS wrote the manuscript with initial input from EJ. DAS created the figures. All authors edited the manuscript.

*Competing interests.*  The authors are unaware of any competing interests.

*Acknowledgements.*  All authors acknowledge support from the UK Natural Environment Research Council (NERC) grant NE/W00531X/1.
DAS further acknowledges support from NERC grant NE/T011920/1.



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
