# Peer review of "FjordRPM v1.0: a reduced-physics model for efficient simulation of glacial fjords"

_EGUsphere, 2024_

## Author Comment (AC2)

**Initial response to reviewer 2**

The reviewer comments are in black, our initial responses are in blue.

In this study, Slater and coauthors developed a reduced-physics model of glacial fjords. This new model is first described and then compared to a 3D high-resolution numerical model. The comparison shows FjordRPM does an excellent job at reproducing the 3D numerical simulation. This model could be used to parameterize glacial fjords in general circulation models, improving the connection between the ocean and the tidewater glaciers/ice sheet.

This study clearly and concisely describes the new model, which is sound, clever and represents a major advance. The evaluation of the model is also well laid out and the model's performance is excellent. I suspect a lot of work behind this well-structured paper and ingenious model. I definitely recommend this manuscript for publication. I have a few minor comments that, I think, could improve this manuscript. These comments are mostly edits, but a few points could be made clearer or justified a little more.

*We thank the reviewer for the time they took to prepare this thorough review – we appreciate it. We are delighted that the manuscript and model have been well received. Below, in the spirit of the interactive discussion, we offer initial responses to the queries and suggestions and indicate how we will adjust the revised manuscript.*

Ln14: "Ocean heat". Thermodynamically "heat" is something that is transferred. While the readers will understand, it might be better to say something like "The ocean heat flux to tidewater glaciers" or define what is meant by heat (thermal energy above the freezing point)?

*Good point – we'll edit as suggested.*

Ln 62-68: Cell volume is V=LWHj : it is a rectangular cuboid, but then it is said that the volume is not the same in all layers. But then that means H changes, but then the sum of H has to equal the fjord's depth. It is also said that the hypsometry of the fjord is taken into account. I am confused, I think this should be clearer.

*Yes – the thickness H_j does not have to be the same in every layer j, but the sum of all the thicknesses has to equal to the fjord depth. In the current version of the model, the fjord is assumed to be a cuboid, meaning that the length L and width W are independent of depth, and the cell volume is V_j=LWH_j. If we allowed for a more flexible fjord hypsometry (not currently implemented), then the length L and width W would become functions of depth, slightly complicating the conversion between layer thickness and volume. We will rewrite this section for clarity.*

Eq. 1: Why not define g' as gΔρ/ρo as is usually done? Moreover, why using a linear equation of state? Matlab was used to code this model, it would be straightforward to call the gsw toolbox to get precise density values, and not computationally expensive either with the polynomial approximation. Over the temperature, pressure and salinity range from your Sermilik example, βT varies by a factor 4 and βS by around 5%. Maybe for version 1.1?

*g', as defined by Eq. 1, is indeed equal to gΔρ/ρo when using a linear equation of state – we can add an intermediate line of algebra to make this clear. As for the use of a linear equation of state – we feel that this is consistent with the overall simplified nature of the model, or put another way, that the error introduced by this choice is probably smaller than that coming from the approximate treatment of other processes (e.g., iceberg melting) or uncertainty in other parameters. But we also see the argument for being precise when it is possible to do so and will take this as a suggestion for future development.*

Ln99: Direct melting of the glacier can also produce plumes (e.g. Zhao et al. 2024), just like for icebergs. I think this section should be clearer that melting along the rest of the glacier is omitted.

*Thank you – yes – we can be clearer on this here in the revision. We recognise the increasing evidence that outside-of-plume melt is a significant portion of the total submarine melt, particularly in winter, and consider this one of the highest priority areas for future development.*

Eq. 21: You use Teff to calculate g'. I am not sure this is right because Teff is 1) not physical (~-85C), and 2) from A3 to A4, where gΔρ ef becomes g'j,melt it says that Δρ ef is the density difference between meltwater and the ambient.

*We think this is ok – the use of an effective temperature in an effective density like this is common in ice-ocean studies – see for example Table 2 of Jenkins (2011) or Eq. (23) of Magorrian & Wells (2016). It accounts for the fact that to melt ice requires latent heat, and this heat must come from the ocean. When thinking about the buoyancy of the meltwater, this latent heat effect can be taken into account by considering this effective density.*

Ln179: "…in mind, but we note that it is not incompatible with regime (ii)": Rotation plays a major role in glacial fjord circulation. In fact, it is mostly a balance between Coriolis and pressure gradients. I think this is worth a little more discussion, either here on in the discussion.

*Agreed – we could have discussed this more and will do so in the revision.*

Ln187: (We) now define?

*Thanks – will change as suggested.*

Figure 1. Add coordinates axes?

*We'll add an indication of the vertical axis here, but the model doesn't – in a formal way – have a horizontal coordinate, so we'd prefer not to add that.*

Ln190: This is only true if there is no barotropic pressure gradient.

*Agreed – we'll note this in the revision.*

Eq. 12-14: This feels like a detour to say the pressure difference is equal to the baroclinic pressure difference: $\Delta p(z) = g \int (\rho s - \rho f jord) dz \ z \ 0$ . I am saying this because I was trying to get to $\rho o \phi$ from (12) and it is kind of circular. Is it hard to keep track of the units and all, starting from (12) to get to $\rho o \phi$. Perhaps even nicer would be to just say Qj Vs is a function of the pressure gradient between the fjord and the coastal shelf, then define pressure (baroclinic and barotropic components) and then approximate the horizontal gradient and introduce your constants? Using volume conservation to estimate ub is definitely right, I just think pressure (baroclinic and barotropic pressure) should be more upfront.

*Ok, yes, we see that starting from the definitions of pressure and going from there would be clearer. Perhaps our current description is a bit upside down. We'll revise this as suggested – thank you.*

Ln195: What is L? I cannot find its value, maybe add to Table A2?

*L is the fjord length (introduced on L64, which is admittedly a while before so we can restate that here). We'll add L (and fjord width W) to Table A1.*

Ln282, Eq 29. KPP is alright and I think this is a good way to go about mixing. However, mixing is super patchy and high mixing only occurs at certain location and time. Therefore, using the same values as Large et al. seems like an overestimation of mixing since it is applied over the whole fjord, likely resulting in (way) too much mixing. Moreover, the area that actually have high mixing (near the glacier and icebergs) indirectly parameterized mixing. There is very few estimates for Kz in the glacial fjord literature, but this section may be worth a citation to Bendtsen et al. 2021. 5x10-3 is high compared their values. Have you tried with smaller values?

[Figure]

*We did try with some different values but felt that, if anything, we might be underestimating mixing (because for example in Fig. 5c-d the FjordRPM temperature and salinity profiles have sharper corners than the MITgcm simulations). Note that 5x10-3 is a maximum value that is*

*only achieved when the Richardson number is negative. Thus, in the simulations this value does only appear in a patchy way – the plot above shows the average FjordRPM Kz in the subglacial discharge simulations in section 4.1.1. High values of vertical mixing are only found around the plume outflow (100 to 300 m depth) and associated with the sill (400 m depth), while at all other depths the vertical mixing is minimal, which we feel is qualitatively consistent with studies such as Bendtsen (to which we'll add a reference). Given the overall good fit of the MITgcm simulations and FjordRPM simulations we'd rather not adjust these values now, but we'll add a note on these points into the revised paper.*

Ln340: It is very fast as is, I think it would be a total loss of time to code this in Fortran, a dying language that no grad student what to learn anymore. If you want to make it faster and open source, I recommend Julia, which you can wrap in python if needed. See oceananigans.jl for example.

*Thanks for these suggestions. Regarding Fortran – we included this as a possibility because it would then be easier to integrate within some earth system models, or for example within MITgcm. But we agree that if using FjordRPM as a standalone model, there would be no sense in converting it into Fortran. We'll clarify this in the text.*

Ln355: what happens if volume is not conserved? Abort or another loop of balancing?

*Good question. The procedure set out in section 2.6 should ensure that volume of the layers is conserved (up to machine precision). We only see problems with volume conservation when the time step is too long, and the model blows up. In such cases, we abort the simulation. We'll clarify this in the revision.*

Section 3.4: I think this section would be better with bullet points

*Good suggestion – we'll make this edit.*

Section 4: It is clear that FjordRPM does a great job at replicating the MITgcm simulation, but could you be more quantitative when comparing both? I am thinking a skill score or at least some $R^2$ in the top corner of plots comparing both. I am assuming you tried a few different vertical discretization or, C0 values, how did you compare between runs?

*Yes – we tried a number of different parameters and chose ones that gave good fits to the MITgcm simulations based just on visual inspection of the plots, rather than on a particular metric. The main sensitivity is to C0, for which we tried a number of values separated by an order of magnitude (C0 = 1E3, 1E4, 1E5, 1E6) before settling on C0 = 1E5 as the value used. We are a bit hesitant to add skill scores to all the plots because we feel that is lending a very precise feel to a comparison that we feel is better viewed as approximate (because MITgcm is not real data or really the truth, and because we haven't optimised the skill scores with a more formal variation of the parameters). But we will add these metrics to the Sermilik simulation plots (Figs. 12 & 13) since that is a more realistic simulation.*

Ln441: ourselves to vary

*Thanks for catching this.*

Figure 3: Please increase the axes label font size.

*We'll increase the size.*

Figure 5: This is impressive. Please increase the font size of the axes label and add salinity units. Caption or label: maybe say panel a) is Qs j/H or similar.

*Thank you – we'll follow these suggestions.*

Ln516: Could this 30% underestimation due to too much vertical mixing? I just still think the minimum and max Kz values are quite high.

[Figure]

*We tested this. In the above plot, case 1 is the same as the manuscript, but with the maximum possible value of vertical diffusivity reduced from 5E-3 to 5E-5. Case 2 is the same as the manuscript but with the background diffusivity reduced to 1E-10. The differences in the exchange volume fluxes are relatively very small – compare the 2E-3 mSv scale of the above plot to Fig. 7a, where the scale reaches 150 mSv. Thus, vertical mixing looks to be playing a very minor role here and can't explain the 30% unfortunately.*

Ln519: Here it says "in general, increasing C0 will strengthen the fjord-shelf exchange" [but not in this case]. What "in general" refers to? The other MITgcm cases?

*"In general" was meant to refer to the fact that C0 linearly scales the fjord-shelf exchange fluxes (Eq. 15), and so without considering any feedbacks, increasing C0 will strengthen the fjord-shelf exchange. And in the other MITgcm cases we found this to be the case. But of course, there is a feedback – increased fjord-shelf exchange will act to dampen the fjord-shelf pressure gradient that also enters Eq. 15. We'll rewrite this sentence to clarify.*

Figure 8: This is remarkable. If not too much of a hassle: make units consistent with the rest of the paper, e.g. m3 s -1 .

*We're not sure exactly what the reviewer means here. In panel (a) we're showing iceberg melt rate and feel that m/d is an appropriate unit since it is very commonly used in the literature. In panel (b) we give the total flux in m3/s, and for the lines in the plot we need to divide the flux by a unit of depth in order to plot it versus depth. We could plot it as m3/s per*

*model layer, but that would then be sensitive to the layer thickness, so we feel that m3/s per metre depth (i.e., m2/s) is the best unit. But if we've misunderstood the comment we can revisit this.*

Ln576: Why 280 m? Also, isn't this deeper than the shallower plume?

*The choice of 280 m was inherited from the MITgcm simulations of Sanchez et al. (2024) – it is the horizontal resolution of their model and the plumes occupied 1 grid cell. We'll clarify this in the manuscript. A depth of 280 m would indeed be deeper than the shallowest plume, but 280 m is the width of the plume (horizontally) and so there is no problem here. For a good illustration, see e.g., Fig. 3a (line plume) of Jackson et al. 2017 – the width of the plume is denoted W in their figure.*

Ln628 and Ln179: You mention you are neglecting an acceleration term in the momentum budget. Could you develop briefly on this, you are also neglecting other terms... Eq. 15 scales the along-fjord volume flux with the along-fjord pressure difference between the fjord and offshore, but did you get there from the actual along-fjord momentum balance or this is an educated that makes sense? Or could you cite someone that shows this scaling makes sense and is neglecting an acceleration term? Or on Ln177, it says "we have derived our exchange ...", could you just put this an appendix like for the iceberg fluxes?

*We got to Eq. 15 from the along-fjord momentum budget and guided by the references provided in that section. Specifically, from the along-fjord momentum budget, we are neglecting both the acceleration term and the momentum advection terms. From there, a shelf exchange in the form of Eq. 15 can be obtained either assuming a balance of pressure gradient and mixing as in Geyer & MacCready (2014), or by a form of geostrophic balance as in Zhao et al. (2021). We appreciate the suggestion to add more explanation and detail here and will do so in the revision.*

Ln657: Bonneau et al. (2024) had an interesting index to describe offshore that englobes process at different timescales. Perhaps it could be used to scale C0?

*Possibly – we would have to think about this more. The index of Bonneau et al. (2024) captures the time variability of the offshore density. We think this is already accounted for in our \phi_j term (Eqs. 14-15) and that the difficulty here lies not in defining the offshore variability but rather in how the exchange responds to that. In L657, we were imagining a parameterisation that would allow for multiple dynamic balances (frictional, geostrophic, hydraulic), somewhat like Eqs. 14-17 of Zhao et al. (2021). But more broadly, we take the point that timescales of variability could enter the problem too and so we will add this possibility and citation to the discussion.*

**References**

Zhao, K. X., Skyllingstad, E. D., & Nash, J. D. (2024). Improved parameterizations of vertical ice-ocean boundary layers and melt rates. Geophysical Research Letters, 51(4), e2023GL105862.

Bendtsen, J., Rysgaard, S., Carlson, D. F., Meire, L., & Sejr, M. K. (2021). Vertical mixing in stratified fjords near tidewater outlet glaciers along Northwest Greenland. Journal of Geophysical Research: Oceans, 126(8), e2020JC016898

Bonneau, J., Laval, B. E., Mueller, D., Hamilton, A. K., & Antropova, Y. (2024). Heat fluxes in a glacial fjord: The role of buoyancy-driven circulation and offshore forcing. Geophysical Research Letters, 51(22), e2024GL111242.

*Jackson, R. H., E. L. Shroyer, J. D. Nash, D. A. Sutherland, D. Carroll, M. J. Fried, G. A. Catania, T. C. Bartholomaus, and L. A. Stearns (2017), Near-glacier surveying of a subglacial discharge plume: Implications for plume parameterizations, Geophys. Res. Lett., 44, 6886–6894, doi:10.1002/2017GL073602.*

*Jenkins, A., 2011: Convection-Driven Melting near the Grounding Lines of Ice Shelves and Tidewater Glaciers. J. Phys. Oceanogr., **41**, 2279–2294, https://doi.org/10.1175/JPO-D-11-03.1.*

*Zhao, K. X., A. L. Stewart, and J. C. McWilliams, 2021: Geometric Constraints on Glacial Fjord–Shelf Exchange. J. Phys. Oceanogr., **51**, 1223–1246, https://doi.org/10.1175/JPO-D-20-0091.1.*

---

## Author Response (AR1)

The reviewer comments are in black, our responses are in blue italics. Line and section numbers refer to the version of the manuscript with changes marked on.

**Reviewer 1**

Summary: I'd be perfectly happy as a reviewer to see this paper published as is. In fact, the reason I'm able to get my review in quickly is because the authors made it so easy to read and review.

The paper starts off with a great name for the new model (FjordRPM). Then comes a tidy and concise schematic in Figure 1 that makes clear what the paper is about. Section 2 cleanly layouts the internals of FjordRPM. It pulls together existing model pieces (like the plume model and aspects of iceberg melt), plus a few new things (like the shelf exchange idealization), into a coherent fjord box model. Section 3 gives a high-level overview of how it's implemented as Matlab code. Then Section 4 proves just how good the model is relative to its efficiency: compared to a full 3D MITgcm model, FjordRPM holds it own. (In fact, as someone who uses the MITgcm for fjord simulations, it's almost a bit disappointing that FjordRPM dynamics are captured so well.)

I can see this model (and future iterations alluded to in the paper) being very useful to folks interested in Greenland-wide scales who cannot simulate the fjords themselves, but want to include their effects. Indeed, the clean and clear Github repo with steps to reproduce all the experiments means it'll be easy for someone to pick this up and configure it for their own use.

Many thanks for taking the time to do this review. We are obviously delighted with this feedback on the manuscript and the surrounding kind words.

It's good practice to use upright text, not italics, in subscripts and superscripts when they are being used as labels: http://physics.nist.gov/cuu/pdf/typefaces.pdf. This is especially useful for improving the look of longer labels like 'eff', 'above', and 'melt'

Thanks for this suggestion – these have been modified throughout.

Consider removing the Conclusion section. The text in the Discussion in more profound and interesting (and less a repeat of the early sections of the paper). It therefore seems, to me at least, to be a much stronger way to finish the paper. Change the heading 'Discussion' to 'Discussion and Conclusion'

Your point about the conclusion is well-taken, but we find ourselves too conventional to do away with this conclusion section and do think there is some value in offering this summary, even if it is by definition a little repetitive.

Typo at 590 'that' > 'than'

Fixed.

**Reviewer 2**

In this study, Slater and coauthors developed a reduced-physics model of glacial fjords. This new model is first described and then compared to a 3D high-resolution numerical model. The comparison shows FjordRPM does an excellent job at reproducing the 3D numerical simulation. This model could be used to parameterize glacial fjords in general circulation models, improving the connection between the ocean and the tidewater glaciers/ice sheet.

This study clearly and concisely describes the new model, which is sound, clever and represents a major advance. The evaluation of the model is also well laid out and the model's performance is excellent. I suspect a lot of work behind this well-structured paper and ingenious model. I definitely recommend this manuscript for publication. I have a few minor comments that, I think, could improve this manuscript. These comments are mostly edits, but a few points could be made clearer or justified a little more.

We thank the reviewer for the time they took to prepare this thorough review – we appreciate it. We are delighted that the manuscript and model have been well received.

Ln14: "Ocean heat". Thermodynamically "heat" is something that is transferred. While the readers will understand, it might be better to say something like "The ocean heat flux to tidewater glaciers" or define what is meant by heat (thermal energy above the freezing point)?

**Edited as suggested (L14).**

Ln 62-68: Cell volume is V=LWHj: it is a rectangular cuboid, but then it is said that the volume is not the same in all layers. But then that means H changes, but then the sum of H has to equal the fjord's depth. It is also said that the hypsometry of the fjord is taken into account. I am confused, I think this should be clearer.

Yes – the thickness H\_j does not have to be the same in every layer j, but the sum of all the thicknesses has to equal to the fjord depth. In the current version of the model, the fjord is assumed to be a cuboid, meaning that the length L and width W are independent of depth, and the cell volume is V\_j=LWH\_j. If we allowed for a more flexible fjord hypsometry (not currently implemented), then the length L and width W would become functions of depth, slightly complicating the conversion between layer thickness and volume. We have rewritten this sentence and removed the reference to fjord hypsometry here as it is an unnecessary source of confusion (L63).

Eq. 1: Why not define g' as  $g\Delta\rho/\rho o$  as is usually done? Moreover, why using a linear equation of state? Matlab was used to code this model, it would be straightforward to call the gsw toolbox to get precise density values, and not computationally expensive either with the polynomial approximation. Over the temperature, pressure and salinity range from your Sermilik example,  $\beta T$  varies by a factor 4 and  $\beta S$  by around 5%. Maybe for version 1.1?

g', as defined by Eq. 1, is indeed equal to  $g\Delta p/po$  when using a linear equation of state – we have added intermediate steps here to make this clear (L91). As for the use of a linear equation of state – we feel that this is consistent with the overall simplified nature of the model, or put another way, that the error introduced by this choice is probably smaller than that coming from the approximate treatment of other processes (e.g., iceberg melting) or uncertainty in other parameters. But we also see the argument for being precise when it is possible to do so and will take this as a suggestion for future development.

Ln99: Direct melting of the glacier can also produce plumes (e.g. Zhao et al. 2024), just like for icebergs. I think this section should be clearer that melting along the rest of the glacier is omitted.

Thank you – yes – we have clarified this point (L109). We recognise the increasing evidence that outside-of-plume melt is a significant portion of the total submarine melt, particularly in winter, and consider this one of the highest priority areas for future development.

Eq. 21: You use Teff to calculate g'. I am not sure this is right because Teff is 1) not physical ( $\sim$ -85C), and 2) from A3 to A4, where g $\Delta$ p ef becomes g'j,melt it says that  $\Delta$ p ef is the density difference between meltwater and the ambient.

We think this is ok – the use of an effective temperature in an effective density like this is common in ice-ocean studies – see for example Table 2 of Jenkins (2011) or Eq. (23) of Magorrian & Wells (2016). It accounts for the fact that to melt ice requires latent heat, and this heat must come from the ocean. When thinking about the buoyancy of the meltwater, this latent heat effect can be taken into account by considering this effective density.

Ln179: "...in mind, but we note that it is not incompatible with regime (ii)": Rotation plays a major role in glacial fjord circulation. In fact, it is mostly a balance between Coriolis and pressure gradients. I think this is worth a little more discussion, either here on in the discussion.

Agreed – we have now rewritten the shelf exchange section (section 2.3) mentioning geostrophic balance (L197) and a possible scaling under a form of geostrophic balance (L208). We've also noted the important of three-dimensional (i.e., including rotational) dynamics in the discussion (L699).

Ln187: (We) now define?

This sentence has been removed in the rewriting of the description of shelf exchange.

Figure 1. Add coordinates axes?

We've added an indication of the vertical axis here, but the model doesn't – in a formal way – have a horizontal coordinate, so we'd prefer not to add that.

Ln190: This is only true if there is no barotropic pressure gradient.

**This sentence has been removed in the rewriting of the description of shelf exchange.**

Eq. 12-14: This feels like a detour to say the pressure difference is equal to the baroclinic pressure difference:  $\Delta p(z) = g \int (\rho s - \rho f j o r d) dz \ z \ 0$ . I am saying this because I was trying to get to  $\rho o \phi$  from (12) and it is kind of circular. Is it hard to keep track of the units and all, starting from (12) to get to  $\rho o \phi$ . Perhaps even nicer would be to just say Qj Vs is a function of the pressure gradient between the fjord and the coastal shelf, then define pressure (baroclinic and barotropic components) and then approximate the horizontal gradient and introduce your constants? Using volume conservation to estimate ub is definitely right, I just think pressure (baroclinic and barotropic pressure) should be more upfront.

Ok, yes, we see that starting from the definitions of pressure and going from there is better. We have revised this section (2.3) as suggested and hope it is now clearer.

Ln195: What is L? I cannot find its value, maybe add to Table A2?

L is the fjord length (introduced on L64, which is admittedly a while before, so we have restated that in L192). We've added L (and fjord width W) to Table A2.

Ln282, Eq 29. KPP is alright and I think this is a good way to go about mixing. However, mixing is super patchy and high mixing only occurs at certain location and time. Therefore, using the same values as Large et al. seems like an overestimation of mixing since it is applied over the whole fjord, likely resulting in (way) too much mixing. Moreover, the area that actually have high mixing (near the glacier and icebergs) indirectly parameterized mixing. There is very few estimates for Kz in the glacial fjord literature, but this section may be worth a citation to Bendtsen et al. 2021. 5x10-3 is high compared their values. Have you tried with smaller values?

We did try with some different values but felt that, if anything, we might be underestimating mixing (because for example in Fig. 5c-d the FjordRPM temperature and salinity profiles have sharper corners than the MITgcm simulations). Note that 5x10-3 is a maximum value that is

only achieved when the Richardson number is negative. Thus, in the simulations this value does only appear in a patchy way – the plot above shows the average FjordRPM Kz in the subglacial discharge simulations in section 4.1.1. High values of vertical mixing are only found around the plume outflow (100 to 300 m depth) and associated with the sill (400 m depth), while at all other depths the vertical mixing is minimal, which we feel is qualitatively consistent with studies such as Bendtsen (to which we'll add a reference). Given the overall good fit of the MITgcm simulations and FjordRPM simulations we'd rather not adjust these values now, but we've added a note on this point into the revised paper (L323).

Ln340: It is very fast as is, I think it would be a total loss of time to code this in Fortran, a dying language that no grad student what to learn anymore. If you want to make it faster and open source, I recommend Julia, which you can wrap in python if needed. See oceananigans.jl for example.

Thanks for these suggestions. Regarding Fortran – we included this as a possibility because it would then be easier to integrate within some earth system models, or for example within MITgcm. But we agree that if using FjordRPM as a standalone model, there would be no sense in converting it into Fortran. This has been clarified in the text (L379).

Ln355: what happens if volume is not conserved? Abort or another loop of balancing?

Good question. The procedure set out in section 2.6 should ensure that volume of the layers is conserved (up to machine precision). We only see problems with volume conservation when the time step is too long, and the model blows up. In such cases, we abort the simulation. This has been clarified in the text (L395).

Section 3.4: I think this section would be better with bullet points

Edited as suggested.

Section 4: It is clear that FjordRPM does a great job at replicating the MITgcm simulation, but could you be more quantitative when comparing both? I am thinking a skill score or at least some R 2 in the top corner of plots comparing both. I am assuming you tried a few different vertical discretization or, CO values, how did you compare between runs?

Yes – we tried a number of different parameters and chose ones that gave good fits to the MITgcm simulations based just on visual inspection of the plots, rather than on a particular metric. The main sensitivity is to CO, for which we tried a number of values separated by an order of magnitude (CO = 1E3, 1E4, 1E5, 1E6) before settling on CO = 1E5 as the value used. We are a bit hesitant to add skill scores to all the plots because we believe that would lend a very precise feel to a comparison that we think is better viewed as approximate (because MITgcm is not real data or really the truth, and because we haven't optimised the skill scores with a more formal variation of the parameters). But we have added mean absolute error statistics to all of the Sermilik simulation plots (Figs. 12 & 13) since that is a more realistic simulation.

Ln441: ourselves to vary

**Thanks for catching this – fixed (L482).**

Figure 3: Please increase the axes label font size.

**Size increased.**

Figure 5: This is impressive. Please increase the font size of the axes label and add salinity units. Caption or label: maybe say panel a) is Qs j/H or similar.

**Edited as suggested.**

Ln516: Could this 30% underestimation due to too much vertical mixing? I just still think the minimum and max Kz values are quite high.

We tested this. In the above plot, case 1 is the same as the manuscript, but with the maximum possible value of vertical diffusivity reduced from 5E-3 to 5E-5. Case 2 is the same as the manuscript but with the background diffusivity reduced to 1E-10. The differences in the exchange volume fluxes are relatively very small — compare the 2E-3 mSv scale of the above plot to Fig. 7a, where the scale reaches 150 mSv. Thus, vertical mixing looks to be playing a very minor role here and can't explain the 30% unfortunately.

Ln519: Here it says "in general, increasing CO will strengthen the fjord-shelf exchange" [but not in this case]. What "in general" refers to? The other MITgcm cases?

"In general" was meant to refer to the fact that C0 linearly scales the fjord-shelf exchange fluxes (Eq. 15), and so without considering any feedbacks, increasing C0 will strengthen the fjord-shelf exchange. And in the other MITgcm cases we found this to be the case. But of course, there is a feedback – increased fjord-shelf exchange will act to dampen the fjord-shelf pressure gradient that also enters Eq. 15. We've rewritten this sentence to clarify (L560).

Figure 8: This is remarkable. If not too much of a hassle: make units consistent with the rest of the paper, e.g. m3 s -1.

We're not sure exactly what the reviewer means here. In panel (a) we're showing iceberg melt rate and feel that m/d is an appropriate unit since it is very commonly used in the literature. In panel (b) we give the total flux in m3/s, and for the lines in the plot we need to

divide the flux by a unit of depth in order to plot it versus depth. We could plot it as m3/s per model layer, but that would then be sensitive to the layer thickness, so we feel that m3/s per metre depth (i.e., m2/s) is the best unit. But if we've misunderstood the comment we can revisit this.

Ln576: Why 280 m? Also, isn't this deeper than the shallower plume?

The choice of 280 m was inherited from the MITgcm simulations of Sanchez et al. (2024) – it is the horizontal resolution of their model and the plumes occupied 1 grid cell. We've clarified this in the manuscript (L619). A depth of 280 m would indeed be deeper than the shallowest plume, but 280 m is the width of the plume (horizontally) and so there is no problem here. For a good illustration, see e.g., Fig. 3a (line plume) of Jackson et al. 2017 – the width of the plume is denoted W in their figure.

Ln628 and Ln179: You mention you are neglecting an acceleration term in the momentum budget. Could you develop briefly on this, you are also neglecting other terms... Eq. 15 scales the along-fjord volume flux with the along-fjord pressure difference between the fjord and offshore, but did you get there from the actual along-fjord momentum balance or this is an educated that makes sense? Or could you cite someone that shows this scaling makes sense and is neglecting an acceleration term? Or on Ln177, it says "we have derived our exchange ...", could you just put this an appendix like for the iceberg fluxes?

We got to Eq. 15 from the along-fjord momentum budget and guided by the references provided in that section. Specifically, from the along-fjord momentum budget, we are neglecting both the acceleration term and the momentum advection terms. From there, a shelf exchange in the form of Eq. 15 can be obtained either assuming a balance of pressure gradient and friction/mixing as in Geyer & MacCready (2014) or Sanchez et al. (2023), or by a form of geostrophic balance as in Zhao et al. (2021). We have rewritten the description of the shelf exchanges (2.3) to provide more detail and make this clearer.

Ln657: Bonneau et al. (2024) had an interesting index to describe offshore that englobes process at different timescales. Perhaps it could be used to scale CO?

Yes – possibly – and this would tie in with the findings of Jackson et al. (2018) who found that the timescale of shelf variability relative to the adjustment timescale of the fjord is an important factor. We have added timescales of variability to the discussion here (L697) and included a reference to Bonneau et al. (2024).

**References**

Zhao, K. X., Skyllingstad, E. D., & Nash, J. D. (2024). Improved parameterizations of vertical ice-ocean boundary layers and melt rates. Geophysical Research Letters, 51(4), e2023GL105862.

Bendtsen, J., Rysgaard, S., Carlson, D. F., Meire, L., & Sejr, M. K. (2021). Vertical mixing in stratified fjords near tidewater outlet glaciers along Northwest Greenland. Journal of Geophysical Research: Oceans, 126(8), e2020JC016898

Bonneau, J., Laval, B. E., Mueller, D., Hamilton, A. K., & Antropova, Y. (2024). Heat fluxes in a glacial fjord: The role of buoyancy-driven circulation and offshore forcing. Geophysical Research Letters, 51(22), e2024GL111242.

Jackson, R. H., E. L. Shroyer, J. D. Nash, D. A. Sutherland, D. Carroll, M. J. Fried, G. A. Catania, T. C. Bartholomaus, and L. A. Stearns (2017), Near-glacier surveying of a subglacial discharge plume: Implications for plume parameterizations, Geophys. Res. Lett., 44, 6886–6894, doi:10.1002/2017GL073602.

Jackson, R. H., Lentz, S. J., and Straneo, F.: The Dynamics of Shelf Forcing in Greenlandic Fjords, Journal of Physical Oceanography, 48,815 2799–2827, https://doi.org/10.1175/JPO-D-18-0057.1, 2018.

Jenkins, A., 2011: Convection-Driven Melting near the Grounding Lines of Ice Shelves and Tidewater Glaciers. J. Phys. Oceanogr., **41**, 2279–2294, https://doi.org/10.1175/JPO-D-11-03.1.

Zhao, K. X., A. L. Stewart, and J. C. McWilliams, 2021: Geometric Constraints on Glacial Fjord—Shelf Exchange. J. Phys. Oceanogr., **51**, 1223—1246, <a href="https://doi.org/10.1175/JPO-D-20-0091.1">https://doi.org/10.1175/JPO-D-20-0091.1</a>.

---

## Author Response (AR2)

Dear Dr. Slater and co-authors:

Thank you for your excellent manuscript, and my apologies for the delay on my end. I was expecting to be in better contact while traveling and in the field; I was wrong.

As you have seen, the referee has just a couple minor comments, which echo my own. Once those are addressed, I would be happy to move forward with your paper.

Sending my best wishes and apologies again for the slow response,

Andy Wickert

Many thanks for handling our paper. We have made the edits suggested by referee #2 and hope that the revised paper will be suitable for publication.

Referee #2

The authors have very well addressed all of my comments. The only minor thing that I think could improve the paper is two small adjustments to the schematic of Figure 1.

- 1) It would be great to show that all layers have the same length. As is, it is ambiguous if the layers follow the bathymetry or not. I am thinking maybe complete the boxes with a dashed line or similar.
- 2) The model can have layers with different height. Maybe it would be nice to have a 2-3 layers with a different height H in the schematic.

Other than that, I would be happy to see this remarkable modeling study published.

Many thanks for the kind words and for taking the time to do the re-review. We have adapted Fig. 1 as suggested.

Kind regards,

Donald Slater
On behalf of the authors